# Systematic review and meta-analyses of studies analysing instructions to authors from 1987 to 2017

Mario Malički 1,2,7✉, Ana Jerončić 3,7, IJsbrand Jan Aalbersberg 4, Lex Bouter 5,6 & Gerben ter Riet 1,2

To gain insight into changes of scholarly journals' recommendations, we conducted a systematic review of studies that analysed journals' Instructions to Authors (ItAs). We summarised results of 153 studies, and meta-analysed how often ItAs addressed: 1) authorship, 2) conflicts of interest, 3) data sharing, 4) ethics approval, 5) funding disclosure, and 6) International Committee of Medical Journal Editors' Uniform Requirements for Manuscripts. For each topic we found large between-study heterogeneity. Here, we show six factors that explained most of that heterogeneity: 1) time (addressing of topics generally increased over time), 2) country (large differences found between countries), 3) database indexation (large differences found between databases), 4) impact factor (topics were more often addressed in highest than in lowest impact factor journals), 5) discipline (topics were more often addressed in Health Sciences than in other disciplines), and 6) sub-discipline (topics were more often addressed in general than in sub-disciplinary journals).

[1] Urban Vitality Centre of Expertise, Amsterdam University of Applied Sciences, Amsterdam, The Netherlands. [2] Amsterdam UMC, University of Amsterdam, Department of Cardiology, Amsterdam, The Netherlands. [3] Department of Research in Biomedicine and Health, University of Split School of Medicine, Split, Croatia. [4] Elsevier, Amsterdam, The Netherlands. [5] Department of Philosophy, Faculty of Humanities, Vrije Universiteit, Amsterdam, The Netherlands. [6] Amsterdam UMC, Vrije Universiteit, Department of Epidemiology and Statistics, Amsterdam, The Netherlands. [7] These authors contributed equally: Mario Malički, Ana Jerončić. ✉email: mario.malicki@mefst.hr

Reporting of research differs between disciplines, and within journals of the same (sub)discipline, on the format and the structure of the manuscript, as well as on the level of detail with which the research is described[1–8]. Instructions to Authors (ItAs) are documents used by journals to describe specific requirements or recommendations authors should follow when reporting research and submitting their manuscript. Additionally, ItAs can describe the type of checks and review processes a journal employs in evaluating received submissions, and how authors or readers can address (suspected) irregularities in published papers[9–12]. ItAs can also be used to promote or raise awareness of standards[13–16], and to depict methods aimed at reducing detrimental research practices, research waste, and the inability to replicate published research[17–19].

Despite 350 years of scholarly publishing, and the existence of >43,000 scholarly journals[20], research on ItAs, and on their evolution and change, is scarce. While it is common practice that journals update their ItAs, the breadth and the extent of changes to ItAs and their variability across disciplines have never been assessed, nor have the insights those changes or differences may provide about the history of scholarly publishing and the development of (best) reporting practices.

Therefore, we synthesised the findings of all studies that have analysed ItAs of more than one journal. After conducting a systematic review, we identified many factors associated with the percentage of ItAs addressing specific topics. Owing to discrepant findings across primary studies, we also conducted a series of meta-analysis to resolve those discrepancies. We focused on the ItAs' recommendations regarding six research integrity topics: authorship, conflicts of interest, data sharing, ethics approval, funding disclosure, and International Committee of Medical Journal Editors Uniform Requirements for Manuscripts (URM).

In this work we summarise 153 studies that analysed ItAs from 1987 to 2017, and we showcase the timeline of ItA changes. We also provide evidence for six factors that explain a substantial part of the wide heterogeneity we found between journals' coverage of the above-mentioned research integrity topics. Those six factors are: (1) time (year when the instructions were applicable), (2) country (in which the journals were published), (3) database (in which the journals were indexed), (4) impact factor, (5) scholarly discipline, and (6) sub-discipline.

## Results

**Study selection and characteristics**. We identified 153 studies eligible for synthesis of results (Fig. 1 - PRISMA Flow Diagram)[21–173]. Study characteristics are shown in detail in Table 1. These studies were published over a thirty-year period (1987 to 2017) with an observed sharp rise in the number of studies following the year 2002 (and that growth was faster than for all articles published in that period, chi-squared tests, $p < 0.0001$ for all comparisons, Fig. 2).

ItAs' contents across these 153 studies were analysed for recommendations or requirements regarding more than a 100 different topics (extracted topics are available in our raw data file)[174]. We grouped those topics into 32 major topics (Table 2), of which the most commonly analysed were Research ethics (i.e., addressing of ethics approvals for conducting studies on humans or animals, $n = 53$, 34%), and Reporting guidelines (i.e., recommendations on items that should be reported for specific research studies, $n = 51$, 33%). The median number of major topics analysed per study was 2 (IQR 1–3). In almost half of the studies ($n = 73$, 48%) researchers also analysed if addressing of a topic was associated with one or more factors, with a total of 15 different factors explored across studies (Supplementary Table 1).

**Narrative synthesis**. We identified 12 different primary objectives authors listed for analysing ItAs (Supplementary Table 2), of which the most common were: (1) to determine if and how a specific topic was addressed in ItAs ($n = 54$, 35%); (2) to determine the reporting or citing of a specific topic in published papers and how the topic was addressed in ItAs ($n = 51$, 33%); (3) to recommend standards for a specific topic ($n = 11$, 7%).

Changes over time were analysed in 11 studies[23,25,26,31,39,43,59–61,67,75,85,91,103,115,134,141,162], covering a time span from 3 to 11 years. Overall, these studies showed that topic coverage increased over time, most notably for: (a) depositing of DNA, amino acid sequence or protein structure data; (b) describing the peer review process; or c) recommending the use of Consolidated Standards of Reporting Trials (CONSORT) Guidelines (Supplementary Table 3).

Differences in reporting of topics in published papers between journals which covered those topics in their ItAs and journals that did not (or at the time when the topics were not addressed) were explored in 17 studies. These mostly showed that reporting is better in journals that covered the topics, albeit suboptimal (i.e., reported in <80% of articles, Supplementary Table 4). Suboptimal adherence to ItAs was also found in 12 studies which analysed if published papers adhered to requirements stated in ItAs (Supplementary Table 5).

**Series of meta-analyses**. We conducted meta-analyses for prevalence of journals covering six research integrity topics: authorship, conflicts of interest, data sharing, ethics approval, funding disclosure, and URM. We chose these six topics due to our interest in research integrity, the project's feasibility, and the number of studies that analysed these topics among the 153 identified studies (Table 1). Reported percentages of journals that covered these topics (with percentages being calculated by dividing the number of journals whose ItAs addressed a topic by the total number of journals whose ItAs were analysed in a particular study) for each individual study are available in our raw data file[174]. For each topic, we found large between-study heterogeneity (i.e., wide ranges of reported percentages, journal sample sizes, and journal selection methods); and in the series of meta-analyses we conducted (see Supplementary Section 2), we found strong effects of 6 factors that explained a substantial part of that heterogeneity, namely: (1) time, (2) country, (3) database indexation, (4) impact factor, (5) discipline, and (6) sub-discipline. However, as more than two-thirds of studies analysed ItAs of Health Sciences journals, these studies dominated the collective evidence. All confirmed effects in the meta-analyses, alongside associations that were reported in individual studies, but which could not be meta-analysed due to how data was reported, are presented in Table 3. Summary results for each factor are presented in subsections below. We chose not to report confidence intervals in the subsections below in order to avoid data overload and to allow for descriptive grouping across topics. However, all results per topic, with associated 95% CIs, are reported in the Supplementary Section 2. Additionally, as time trends were estimated using regression models, percentages reported below may differ from the percentages reported in individual studies.

**Time**. We found large differences between 1986 and 2016 in percentages of ItAs addressing the six above-mentioned research integrity topics. Overall topic coverage generally increased over time. For example, while in 1995, ~40% of top or Abridged Index Medicus Health Sciences journals addressed authorship and ethics approval, by 2005, >70% of those journals did so (Fig. 3 and Supplementary Information). In the same period, a similar

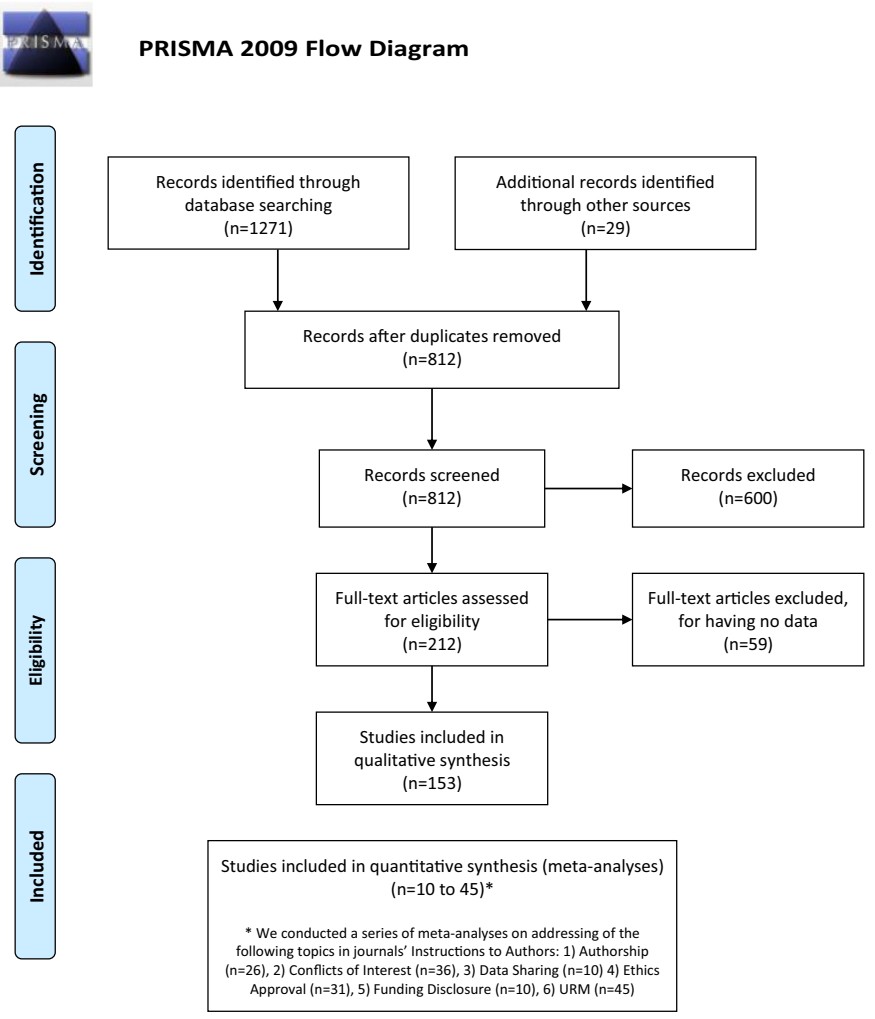

**Fig. 1 PRISMA flow diagram.** After screening 812 records, we synthesized 153 studies analysing journals' Instructions to Authors (ItAs).

increase was found in UK and USA Health Sciences journals for ethics approval, while Indian and Brazilian Health Sciences journals experienced an increase a decade later (Indian journals, from 57% in 2004 to 78% in 2015, and Brazilian journals, from 56% in 2007 to 83% in 2012).

An increase over time, however, was not ubiquitous for all topics, (sub)disciplines or countries. In Health Sciences sub-disciplinary journals, addressing of conflicts of interest increased from 57% in 1995 to 87% in 2015, but those journals showed no increase after 2000 for URM (60%), authorship (65%), or funding disclosure (81%).

We also observed a decrease in addressing of URM in Abridged Index Medicus Health Sciences journals, from 35% in 1986 to 5% in 2006.

Data on changes over time for non-Health Sciences journals was scarce. Top journals (of all disciplines) showed an increase for data sharing (from 15% in 1992), and for conflicts of interest (from 16% in 1997), to >85% for both topics by the year 2010 (Fig. 3). Additionally, we found an indication of an increase over time for Croatian journals (from all disciplines) for authorship, conflicts of interest, and data sharing from ~9% in 2013 or 2014 to >20% in 2015 (Supplementary Information).

**Country**. We found large differences in addressing of topics between countries. Almost always, topic coverage was lower in journals from a single country than among top or general Health

Sciences journals. For example, while in 2010 ~83% of top journals (of all disciplines) addressed conflicts of interest, in 2014, 89% of Indian Health Sciences journals did so, and only 9% of Croatian journals. Similarly, while in 2014 almost 90% of top Health Sciences journals addressed authorship, 86% of Chinese journals did so, 70% of Indian, and 29% of Croatian journals.

Among Health Sciences journals, addressing of conflicts of interest (89% in 2014), funding disclosure (70% in 2008), and URM (75% in 2014) was most prevalent in Indian journals, ethics approval (86% in 2005) in UK journals, and authorship (86% in 2014) in Chinese journals. Chinese journals, however, had the lowest coverage of URM (7% in 2011).

Country-specific data for journals of all disciplines was only available for Cameroon in 2009, and Croatia for periods 2012 to 2015, with coverage of all topics found in <37% of journals (Supplementary Information).

**Journal indexation**. Journal indexation was associated with covering of all topics except data sharing (for which no studies provided data for journals from different databases). For example, in 1986, higher percentage of top Health Sciences journals addressed funding disclosure than did Abridged Index Medicus journals (47% vs. 22%, respectively). That situation was reversed for ethics approval 20 years later: with 81% of Abridged Index Medicus journals in 2006 vs. 71% of top Health Sciences journals in 2009 (Supplementary Information). Additionally, while Health

**Table 1 Characteristics of studies that analysed Instructions to Authors (ItAs) included in the systematic review, as well as those meta-analysed per specific topic.**

| | Systematic review (n = 153) | Meta-analyses | | | | | |
|---|---|---|---|---|---|---|---|
| | | Authorship (n = 26) | Conflicts of interest (n = 36) | Data sharing (n = 10) | Ethics approval (n = 31) | Funding disclosure (n = 10) | ICMJE URM[a] (n = 45) |
| Year of study publications (range) | 1987–2017 | 1999–2017 | 1987–2017 | 1995–2016 | 1997–2017 | 1987–2016 | 1987–2017 |
| Year of ItAs that were analysed (range) | 1976–2016 | 1995–2015 | 1986–2015 | 1992–2015 | 1995–2015 | 1976–2015 | 1986–2016 |
| No (%) of publications not listing the ItA year information | 69 (45) | 9 (35) | 12 (33) | 4 (40) | 12 (39) | 4 (36) | 11 (24) |
| ItAs analysed per publication (median, range) | 56 (3–1396) | 57 (5–445) | 54 (5–1396) | 60 (5–850) | 65 (4–208) | 68 (6–216) | 95 (4–747) |
| Discipline analysed (n, %) | | | | | | | |
| Arts and humanities | 0 (0) | 0 (0) | 0 (0) | 0 (0) | 0 (0) | 0 (0) | 38 (84) |
| Health sciences | 116 (76) | 18 (69) | 29 (81) | 2 (20) | 26 (84) | 7 (64) | 0 (0) |
| Life sciences | 3 (2) | 0 (0) | 0 (0) | 1 (10) | 0 (0) | 0 (0) | 0 (0) |
| Physical sciences | 3 (2) | 0 (0) | 0 (0) | 0 (0) | 0 (0) | 0 (0) | 0 (0) |
| Social sciences | 7 (5) | 1 (4) | 0 (0) | 0 (0) | 0 (0) | 0 (0) | 1 (3) |
| Multiple | 24 (16) | 7 (27) | 9 (19) | 7 (70) | 5 (16) | 4 (36) | 6 (13) |
| Countries/regions of journals analysed (n, %) | | | | | | | |
| Multiple | 106 (69) | 11 (42) | 19 (53) | 8 (80) | 11 (35) | 6 (55) | 24 (53) |
| Brazil | 12 (8) | 1 (4) | 4 (11) | 0 (0) | 8 (26) | 1 (9) | 5 (11) |
| India | 6 (4) | 3 (12) | 2 (6) | 0 (0) | 6 (19) | 1 (9) | 4 (9) |
| China | 5 (3) | 1 (4) | 1 (3) | 0 (0) | 0 (0) | 0 (0) | 1 (2) |
| Croatia | 5 (3) | 4 (15) | 2 (6) | 2 (20) | 1 (3) | 1 (9) | 3 (7) |
| Spain | 3 (2) | 1 (4) | 1 (3) | 0 (0) | 0 (0) | 0 (0) | 1 (2) |
| Mexico | 2 (1) | 0 (0) | 1 (3) | 0 (0) | 0 (0) | 0 (0) | 0 (0) |
| South Korea | 2 (1) | 0 (0) | 0 (0) | 0 (0) | 1 (3) | 0 (0) | 2 (4) |
| Other (1 per country)[b] | 12 (8) | 5 (20) | 6 (17) | 0 (0) | 4 (13) | 2 (18) | 5 (11) |
| Journal selection methods (n, %) | | | | | | | |
| All journals within a database | 88 (57) | 23 (88) | 28 (78) | 6 (60) | 25 (81) | 8 (73) | 32 (71) |
| No. of top journals within a database | 35 (23) | 0 (0) | 2 (6) | 3 (30) | 1 (3) | 0 (0) | 9 (20) |
| Random sample of journals | 1 (1) | 0 (0) | 0 (0) | 0 (0) | 0 (0) | 0 (0) | 0 (0) |
| All journals with an impact factor >10 | 1 (1) | 0 (0) | 0 (0) | 0 (0) | 0 (0) | 0 (0) | 0 (0) |
| All journals with an impact factor >2.63 | 1 (1) | 0 (0) | 0 (0) | 0 (0) | 0 (0) | 0 (0) | 0 (0) |
| Authors' choice | 9 (6) | 1 (4) | 3 (8) | 1 (10) | 1 (3) | 1 (9) | 0 (0) |
| A combination of methods | 10 (7) | 2 (8) | 2 (6) | 0 (0) | 1 (3) | 2 (18) | 2 (4) |
| Selection method not listed | 8 (5) | 0 (0) | 1 (3) | 0 (0) | 3 (10) | 0 (0) | 2 (4) |
| Analytic method (n, %) | | | | | | | |
| Not specified | 94 (61) | 12 (46) | 19 (53) | 5 (50) | 21 (68) | 5 (45) | 20 (44) |
| Two independent coders | 32 (21) | 7 (27) | 12 (33) | 1 (10) | 8 (26) | 3 (27) | 15 (33) |
| One coder | 15 (10) | 3 (11) | 1 (3) | 2 (20) | 2 (6) | 2 (18) | 7 (16) |
| One author extracted the data, the other checked | 6 (4) | 2 (8) | 1 (3) | 1 (10) | 0 (0) | 0 (0) | 1 (2) |
| One coder extracted data, the other checked, third checked a random sample | 2 (1) | 1 (4) | 1 (3) | 0 (0) | 0 (0) | 1 (9) | 1 (2) |
| Two coders independently assessed a portion of the sample, then proceeded independently to extract from the remaining journals | 2 (1) | 0 (0) | 1 (3) | 0 (0) | 0 (0) | 0 (0) | 0 (0) |
| One coder plus help of a text mining software | 1 (1) | 1 (4) | 0 (0) | 1 (10) | 0 (0) | 0 (0) | 0 (0) |
| One author extracted sentences related to the topics analysed, then two proceeded to code the extracted sentences | 1 (1) | 0 (0) | 1 (3) | 0 (0) | 0 (0) | 0 (0) | 1 (2) |

[a]International Committee of Medical Journal Editors (ICMJE) Uniform Requirements for Manuscripts Submitted to Biomedical Journals (URM).
[b]Germany; Pakistan; Latin America and Caribbean; Spain and Latin American countries; Brazil, Mexico, Argentina, and Chile; Countries of Eastern and Southern Europe; Hungary and bordering countries; Japan; Cameroon; Taiwan; India and UK; Iran.

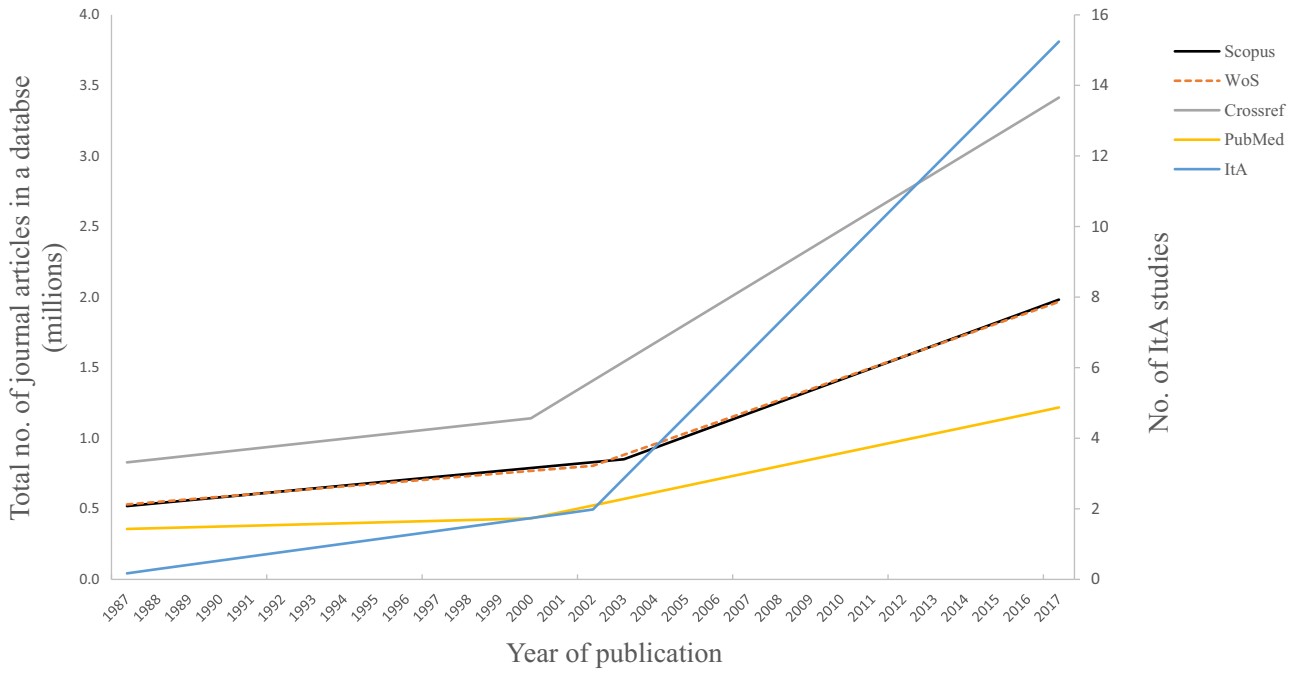

**Fig. 2 Growth of the number of publications analysing journals' Instructions to Authors (ItAs).** Growth of ItA studies is shown alongside that of journal articles in Crossref, PubMed, Scopus, and Web of Science. Prediction lines were determined by optimal spline regression models.

**Table 2 Number of publications analysing a specific topic in journals' Instructions to Authors ($N = 153$).**

| Topic | n | % | Topic | n | % |
|---|---|---|---|---|---|
| Research ethics | 53 | (34) | Journal's scope | 6 | (3) |
| Reporting guidelines | 51 | (33) | Publication of supplementary materials | 6 | (3) |
| ICMJE URM | 44 | (28) | Abbreviations | 5 | (3) |
| Conflicts of interest | 36 | (23) | Acknowledgments | 5 | (3) |
| Authorship | 35 | (22) | Addressing of sex or ethnicity | 5 | (3) |
| Clinical trial registration | 27 | (17) | Editorial policies | 4 | (2) |
| Publication ethics | 26 | (16) | Journal financial disclosure | 3 | (1) |
| Manuscript formatting | 20 | (13) | Journal self-archiving policy | 3 | (1) |
| Accepted article type | 15 | (9) | Text legibility | 2 | (1) |
| Peer review | 14 | (9) | Registration of systematic reviews | 2 | (1) |
| Referencing | 12 | (7) | Submission format (e.g., email or print) | 2 | (1) |
| Copyright policy | 11 | (7) | Cover letter | 1 | (0) |
| Data sharing | 11 | (7) | Editorial freedom | 1 | (0) |
| Funding disclosure | 10 | (6) | Manuscript limitations | 1 | (0) |
| Statistics | 7 | (4) | Use of medical subject headings | 1 | (0) |
| Committee on publication ethics | 6 | (3) | Replication | 1 | (0) |

Sciences journals indexed in Journal Citation Reports (JCR) showed an increase in URM coverage from 2001 (22%) to 2014 (77%), Abridged Index Medicus Health Sciences journals showed a decrease between 1986 (37%) and 2006 (5%).

For Health Sciences sub-disciplinary journals, almost no differences were found between journals indexed in Directory of Open Access Journals (DOAJ), Index Medicus (IM) or JCR between 2008 and 2016 for URM (60%), however paediatrics journals index in JCR addressed conflicts of interest (78%) more often than DOAJ indexed journals (63%).

**Impact factor**. We found weak indications that the coverage of authorship, conflicts of interest, and URM was associated with impact factor for Health Sciences sub-disciplinary journals (no data was available for other disciplines, or even for general Health Sciences journals). Specifically, for conflicts of interest, higher coverage was found for IF ≥ 3 journals (85%) compared to

journals with IF < 1 (72%) between 2008 and 2016, as well as for URM (74% vs. 50%). For authorship, higher coverage was found in 2010 for journals with IF values of 1 to 2 (61%) than for those with IF < 1 (26%). Single studies, and studies reporting correlation analyses with IF yielded inconclusive evidence (see additional analyses below and the Supplementary Information).

**Discipline**. We found large disciplinary differences for all topics, with Health Science journals more often addressing all six research integrity topics (e.g., in 2010 in Web of Science, 59% of Health Sciences journals addressed authorship vs. 7% of Arts & Humanities journals).

However, only 1–3 studies per topic reported disciplinary data, and those were either based on data from 2010 onwards or belonged to country or region-specific disciplinary journals (Croatia, Spain and Latin America, or Spain and the Caribbean, Supplementary Information).

**Table 3 Factors associated with addressing of topics in journals' Instructions to Authors (ItAs). Factors were confirmed by meta-regression or by demonstrating significant differences when data were obtained from up to three studies** (white background)**, calculated from data reported in individual studies** (green background) **or reported as presented in individual studies, i.e., data reported in a way that did not allow calculation** (grey background)**. All associations are presented only descriptively, with full details and 95% CIs available in the Supplementary Information. We used additional colouring to highlight** disciplines (blue)**,** countries (red)**, and** other categories (purple)**.**

| Topic | Time | Country | Indexation | Impact factor (IF) | Discipline† | Subdiscipline† | Other |
|---|---|---|---|---|---|---|---|
| | | | | **Global findings** | | | |
| **Authorship**<br><br>Authorship percentage (AP) is defined as a number of journals whose Instructions to Authors addressed authorship, divided by the total number of journals analysed in a study. Several APs were sometimes reported per study for different journal sub-groups or time periods. | Sig. increase over time for top and AIM indexed Health Sciences journals between 1995 (46%) and 2015 (86%).<br><br>No sig. change for Health Sciences sub-disciplinary journals between 1995 and 2015 (summary AP 52%).<br><br>No change for nine general Health Sciences journals between 2000 and 2010 (5 out of 9 journals addressed authorship). | High heterogeneity between Health Sciences journals of eight different countries (e.g. in 2014, reported AP for Croatia was 14%, for India 70%, and for China 85%). | Indirect evidence: annual increase for top and AIM indexed Health Sciences was attenuated when IM indexed journals were included in the meta-regression model.<br><br>No sig. differences in 2013 between MEDLINE indexed Health Sciences journals and non-indexed journals (summary AP 52%), nor between PMC indexed and non-indexed journals (summary AP 52%). | Higher AP for Health Sciences sub-disciplinary (otorhinolaryngology) journals in 2010 with IF values of 1 to 2 (61%) compared to those with IF < 1 (26%). Inconsistent result for journals of IF > 2 category. (Possibly due to the small sample size in that category - 6 journals.)<br><br>Higher APs for general medicine Health Sciences journals with IF > 2.2 (3 out of 3 analysed journals) vs IF < 1 (0 out of 3 journals). Sample size too small to determine differences for IF 1 to 2 category (2 out of 3 journals). Same journals were analysed in 2000, 2005 and 2010 and showed no changes over time. | Lowest AP was found in Arts & Humanities journals (7%) in 2010 (38% to 80% for other disciplines). | Lower APs of Health Sciences sub-disciplinary journals compared to general and top Health Sciences journals.<br><br>No sig. change for Health Sciences sub-disciplinary journals between 1995 and 2015 (summary AP 52%). | Sig. difference in 2006 between WAME member Health Sciences journals (70%) and non-member journals (40%).<br><br>No sig. differences in 2013 between ICMJE endorsing Health Sciences pharmacy journals and non-endorsing journals (summary AP 51%). |
| | | | | **Country specific findings** | | | |
| | No sig. change for Indian Health Sciences journals between 2010 and 2015 (summary AP 65%).<br><br>No sig. change for Croatian Health Sciences journals between 2014 (29%) and 2015 (37%).<br><br>No sig. change for Croatian journals (across all disciplines) between 2013 (9%), 2014 (14%) and 2015 (18%). | Higher AP of Spanish and Latin America Social Sciences journals (25% in 2015), compared Croatian Social Sciences journals between 2013 and 2015 (summary AP 10%). | | No sig. association between impact factor and addressing of authorship for Indian Health Sciences journals in 2014. | Higher AP for Croatian Health Sciences journals (29%) in 2014 compared to non-Health journals (11%).<br><br>Higher AP for Croatian Health Sciences journals (37%) in 2015 compared to Arts & Humanities, Social Sciences, or Technical Sciences (8% to 29 %). | | Higher AP of Iranian Health Sciences journals in 2012 publishing in English (25%) compared to those publishing in Farsi (6%).<br><br>No sig. difference in 2014 between Chinese CMPAH Health Sciences journals and non-CMPAH journals (summary AP 85%). |
| | | | | **Global findings** | | | |
| **Conflicts of interest**<br><br>Conflicts of interest percentage (CP) is defined as a number of journals whose Instructions to Authors addressed conflicts of interest, divided by the total number of journals analysed in a study. Several CPs were sometimes reported per study for different journal sub-groups or time periods. | Sig. increase over time across disciplines between 1997 (16%) and 2010 (100%).<br>(Possibly not a real trend, but an effect of the smaller sample size of latter two studies, n=41 in 1998, and n=5 in 2010, compared to the first study, n=1396 in 1997, as well as the higher ratio of Health Science journals included in the latter two studies – see effect of discipline.)<br><br>Sig. increase over time for AIM indexed Health Sciences between 1986 (10%) and 2005 (95%).<br><br>Sig. increase over time for top Health Sciences journals between 1986 (21%) and 2010 (83%).<br><br>Sig. increase over time for Health Sciences sub-disciplinary journals (clinical specialties) index in JCR between 1995 (64%) and 2015 (96%). | High heterogeneity between Health Sciences journals of eight different countries or regions (e.g., in 2014, reported CP for Croatia was 32% and for India 89%). | Higher CP of Health Sciences paediatrics journals index in JCR (78%) compared to those indexed in DOAJ (63%).<br><br>No sig. differences in 2013 between MEDLINE indexed Health Sciences pharmacy journals and non-indexed journals (summary CP 43%), nor between PMC indexed and non-indexed journals (summary CP 43%). | No sig. difference between top 30 Health Sciences journals in 2009 with IF ≥ 10 and 30 journals with IF < 10 (summary CP 100%).<br>(Possibly due to high IF values of the other group too.)<br><br>Increase in CP for Health Sciences journals (general medicine) for journals with IF < 1 (1 out of 3 journals in 2000 to 2 out of 3 journals in 2005 and 2010). No difference was observed for IF 1 to 2 (2 out of 3 journals) or IF > 2.2 (3 out of 3 journals) between 2000 and 2010.<br><br>Sig. difference between IF > 3 (85%) and IF < 1 (72%) Health Sciences sub-disciplinary journals from 2008 to 2015, with no sig. difference in comparison with journals with IF 1 to 2, and IF 2 to 3. | Higher CPs of Health Sciences journals in 1995 (53%) and 2001 (75%) compared to top journals across disciplines in closest available timepoints 1997 (16%) and 1998 (41%). | Sig. differences between Health Sciences sub-disciplinary journals, with journals of clinical specialties from 1995 to 2015 (summary CP 79%), or general medicine from 2000 to 2010 (summary CP 74%) having higher CPs compared to those of pharmacy journals in 2013 (42%). | Sig. difference in 2009 between Health Sciences paediatrics journals of open access publishing houses (100%), professional organisation publishers (50%), and other publishers (43%).<br><br>No sig. differences in 2013 between ICMJE endorsing Health Sciences pharmacy journals and non-endorsing journals (summary AP 43%). |
| | | | | **Country specific findings** | | | |
| | Sig. increase for Croatian journals (across all disciplines) between 2014 (9%) and 2015 (21%).<br><br>Non-sig. increase for Croatian Health Sciences journals between 2014 (32%) and 2015 (43%).<br><br>Non-sig. increase for Brazilian Health Sciences journals between 2007 (55%) and 2012 (73%).<br><br>Sig. increase for Indian Health Sciences journals between 2008 (30%) and 2014 (89%). | No sig. difference between Cameroonian journals (across all disciplines) in 2009 (22%) and Croatian journals in 2013 (9%) or 2015 (21%).<br>(Possible lack of effect due to only 9 Cameroonian journals analysed, compared to 197 in 2014, and 283 in 2015 for Croatia.)<br><br>No sig. difference in 2015 between Croatian Social Sciences journals (24%) and Spanish and Latin American journals (23%). | | Higher CP of Brazilian Health Sciences journals in 2012 with IF (92%) vs journals without IF values (54%).<br><br>No sig. association between impact factor and addressing of conflicts of interest for Indian Health Sciences journals in 2014. | Higher CP for Croatian Health Sciences journals (32%) in 2014 compared to non-Health journals (4%).<br><br>No sig. difference in 2015 between Croatian Social Sciences journals (24%) and Spanish and Latin American journals (23%). | | No sig. difference in 1995 between ICMJE endorsing Mexican Health Sciences journals and non-endorsing journals (summary CP 7%).<br><br>Higher CP of Iranian Health Sciences journals in 2012 publishing in English (79%) compared to those publishing in Farsi (31%).<br><br>Sig. difference in 2014 between Chinese CMPAH Health Sciences journals (34%) and non-CMPAH journals (6%). |
| | | | | **Global findings** | | | |
| **Data sharing**<br><br>Data sharing percentage (DP) is defined as a number of journals whose Instructions to Authors addressed data sharing, divided by the total number of journals analysed in a study. Several DPs were sometimes reported per study for different journal sub-groups or time periods. | Sig. increase over time for top journals across disciplines between 1992 (15%) and 2010 (88%).<br><br>No sig. change over time for sub-disciplinary journals of different disciplines (DP of 74% for molecular biology and biochemistry in 2007, 68% for substance abuse in 2013, and 20% for biodiversity conservation in 2015) (2015 study only analysed 5 journals). | Possible indirect country effect, as Croatian journals had lower DPs then were found in top journals across disciplines, however, this effect could be due to the influence of top journals or impact factor. | Sig. association of open access journals and existence of the data sharing policy for gene expression microarray data in 2006. | Sig. increase in DPs of Social Sciences sub-disciplinary journals (substance abuse) in 2013 per IF quartile category (22% increase per quartile, with DP for Q1 being 38%, and for Q4 89%).<br><br>Sig. association of impact factor of journals publishing gene expression profiling and existence of a gene expression microarray data sharing policy in 2006.<br><br>Sig. association of impact factor and mandatory provision of materials and protocols in 2009 for top journals across disciplines. | | Sig. neg. association of Oncology journals and data sharing policy for gene expression microarray data in 2006.<br><br>No sig. change over time for sub-disciplinary journals of different disciplines (DP of 74% for molecular biology and biochemistry in 2007, 68% for substance abuse in 2013, and 20% for biodiversity conservation in 2015) (2015 study only analysed 5 journals). | |
| | | | | **Country specific findings** | | | |
| | Sig. increase for Croatian journals (across all disciplines) between 2014 (9%) and 2015 (36%). | | | | Higher DP for Croatian Health Sciences journals (24%) in 2014 compared to non-Health journals (6%).<br><br>Lower DP for Croatian Arts & Humanities journals (19%) in 2015 compared to Natural (62%) and Biotechnical Sciences (77%). | | |

**Sub-discipline.** Generally, topics were less often addressed in Health Sciences sub-disciplinary than in top or general Health Sciences journals. For authorship, funding disclosure, and URM there were almost no sub-disciplinary differences (52% for authorship between 1995 and 2015, 81% for funding disclosure between 2000 and 2015, and 60% for URM between 2008 and 2016, Fig. 3). However, we found large differences for ethics approval and conflicts of interest between sub-disciplinary

## Table 3 (continued)

| Topic | Time | Country | Indexation | Impact factor (IF) | Discipline† | Subdiscipline† | Other |
|---|---|---|---|---|---|---|---|
| **Ethics approval** — Ethics approval percentage (EP) is defined as a number of journals whose Instructions to Authors addressed ethics approval, divided by the total number of journals analysed in a study. Several EPs were sometimes reported per study for different journal sub-groups or time periods. | *Global findings* — Sig. increase over time for Health Sciences journals between 1995 and 2009, with an additional effect of indexation: higher EP were found in journals indexed in AIM (max 81% reached in 2006) compared to top journals (max of 72% reached in 2009). | High heterogeneity between Health Sciences journals of seven different countries and two regions (e.g., in 2012 reported EP for Southeast European countries was 41% and in 2010 for Brazil 77%). | Higher EP for AIM indexed Health Sciences journals (max EP of 83% reached in 2006) compared to top journals (max EP of 71% reached in 2009). | Increase in EP for general medicine Health Sciences journals for journals with IF < 1 (1 out of 3 journals in 2000 and 2005 to 2 out of 3 journals in 2010). No difference was observed for IF 1 to 2 (0 out of 3 analysed journals) or IF > 2.2 (3 out of 3) between 2000 and 2010. No sig. association of impact factor and ethics approval in 2009 between top 30 Health Sciences journals with IF > 10 and 30 with IF < 10. Possibly due to also high IF in the latter group. No sig. association of impact factor and ethics approval in 2015 Health Sciences sub-disciplinary journals (medical laboratory technology). | | Sig. differences between Health Sciences sub-disciplinary journals with an additional effect of time (e.g., internal medicine journals had EP of 61% in 1995, and dentistry journals in 2010 had 45%). | |
| | *Country specific findings* — Sig. increase over time for Indian Health Sciences journals between 2004 (57%) and 2015 (78%). Sig. increase over time for Brazilian Health Sciences journals between 2007 (56%) and 2012 (83%). Sig. increase over time for UK Health Sciences journals between 1995 (43%) and 2005 (86%). Sig. increase over time for USA Health Sciences journals between 1995 (42%) and 2005 (75%). | No sig. difference between Cameroonian journals (across all disciplines) in 2009 (22%) and Croatian journals in 2014 (8%). (Possibly due to the small number of Cameroonian journals analysed, n=9, compared to the number of Croatian journals analysed, n=197). | Higher EP for JCR indexed Brazilian Health Sciences journals in 2012 that had an IF value (88%) compared to those sampled from Webqualis/CAPES that did not have an IF value (67%). No sig. difference in 2005 and 2008 between MEDLINE indexed Indian Health Sciences journals and non-indexed journals (summary EP 55% in 2005, and 70% in 2008). | Higher EP for Brazilian Health Sciences journals in 2012 with IF values (88%) vs journals without IF values (67%). Sig. association of impact factor and ethics approval of UK, USA and Canadian Health Sciences journals in 2005. No sig. association between impact factor and ethics approval for Indian Health Sciences journals in 2014. | Higher EP for Croatian Health Sciences journals (21%) in 2014 compared to non-Health journals (5%). | | No sig. difference in 1997 between South Korean Health Sciences Medical Association member journals and quasi-member journals (summary EP of 4%). Sig. difference in 2005 and 2008 between ICMJE endorsing Indian Health Sciences journals (79% in 2005 and 91% in 2008) and non-endorsing journals (41% in 2005 and 61% in 2008). |
| **Funding disclosure** — Funding disclosure percentage (FP) is defined as a number of journals whose Instructions to Authors addressed funding disclosure, divided by the total number of journals analysed in a study. Several FPs were sometimes reported per study for different journal sub-groups or time periods. | *Global findings* — No sig. change over time for top Health Sciences journals for between 1986 and 1998 (summary FP of 47%). No sig. change over time for Health Sciences sub-disciplinary journals between 2000 and 2015 (summary FP of 81%). | Higher FP for Indian Health Sciences in 2008 (70%) compared to Southeast European Health Sciences journals in 2012 (29%) or to Croatian Health Sciences journals in 2014 (37%). | Higher FPs in 1986 for top Health Sciences journals (40%) or AIM indexed journals (22%) compared to IM indexed (3%) or non-indexed journals (1%). Higher FP for top Health Sciences journals (summary FP 47% between 1986 and 1998) compared to AIM indexed journals (22% in 1986). | Decrease in FP for Health Sciences journals (general medicine) for journals with IF < 1 (3 out of 3 journals in 2000 and 2005, to 2 out of 3 journals in 2010). No difference was observed for IF 1 to 2 (0 out of 3 journals) or IF > 2.2 (3 out of 3 journals) between 2000 and 2010. | Lower FPs for Croatian non-Health Sciences journals (12%) and for and Spanish and Latin American Social Sciences journals (6%) compared to FPs of Health Sciences journals reported in nine studies (28% to 93%). | No sig. differences for Health Sciences sub-disciplinary journals between 2000 and 2015 (summary FP of 81%). | |
| | *Country specific findings* | | | | Higher FP for Croatian Health Sciences journals (37%) in 2014 compared to non-Health journals (12%). | | |
| **ICMJE's URM** — International Committee of Medical Journal Editors (ICMJE) Uniform Requirements for Manuscripts Submitted to Biomedical Journals (URM) percentage (UP) is defined as a number of journals whose Instructions to Authors addressed ICMJE's URM, divided by the total number of journals analysed in a study. Several UPs were sometimes reported per study for different journal sub-groups or time periods. | *Global findings* — Sig. increase over time for JCR indexed Health Sciences journals between 2001 (26%) and 2014 (76%). Sig. decrease over time for AIM indexed Health Sciences between 1986 (37%) and 2006 (5%). No sig. differences for Health Sciences sub-disciplinary journals indexed in DOAJ, IM or JCR between 2008 and 2016 (summary UP of 60%). | No sig. differences between top 5 Health or Life Sciences journals of four countries (Brazil, Mexico, Chile and Argentina) in 2013 (UPs of 0 to 40%). (High uncertainty of estimates due to small sample sizes) High heterogeneity between Health Sciences journals of different countries (e.g., in 2011 reported UP for Chinese journals was 7%, and in 2012 for Latin American and Caribbean journals 61%). No sig. association with publication country and Health Sciences paediatrics journals UPs in 2010. | Higher UPs in 1986 for top Health Sciences journals (53%) or AIM indexed journals (37%) compared to IM indexed (11%) or non-indexed journals (6%). Indirect evidence due to sig. increase for JCR indexed Health Sciences journals between 2001 (26%) and 2014 (76%), while AIM indexed journals had a sig. decrease between 1986 (37%) and 2006 (5%). No sig. differences for Health Sciences sub-disciplinary journals indexed in DOAJ, IM or JCR between 2008 and 2016 (summary UP of 60%). Sig. differences in 2013 between MEDLINE indexed Health Sciences pharmacy journals (60%) and non-indexed journals (83%), with no. sig. differences between PMC indexed and non-indexed journals (summary UP 73%). | Sig. difference between Health Sciences sub-disciplinary journals with IF ≥ 3 compared to those with IF < 1. No differences in relation to middle (IF ≥ 1 to 2 or IF ≥ 2 to 3) categories. Indirect evidence due to higher UP of top 15 JCR indexed general medicine Health Sciences journals in 2003 (87%) vs top 151 clinical specialties (22%). Indirect evidence due to higher UP of top 15 Health Sciences journals in 1986 (53%) vs top 124 JCR indexed Health Sciences journals in 2001 (22%). | Higher UP of top 15 JCR indexed general medicine Health Sciences journals in 2003 (87%) vs top 151 clinical specialties (22%). | Higher UP of top 15 JCR indexed general medicine Health Sciences journals in 2003 (87%) vs top 151 clinical specialties (22%). No sig. differences for Health Sciences sub-disciplinary journals indexed in DOAJ, IM or JCR between 2008 and 2016 (summary UP of 60%). | No sig. difference in 2001 between ICMJE endorsing Health Sciences journals and CONSORT endorsing journals (summary UP of 71%). Sig. difference in 2003 between CONSORT endorsing Health Sciences journals (72%) and non-endorsing journals (35%). No sig. difference in 2009 between Health Sciences paediatrics journals of open access publishing houses, professional organisation publishers or other publishers (summary UP of 65%). No sig. association in 2010 of medical association membership or publication language for Health Sciences paediatrics journals and URM. Sig. differences in 2013 between ICMJE endorsing Health Sciences pharmacy journals (93%) and non-endorsing journals (58%). |
| | *Country specific findings* — No sig. change over time for Croatian journals (across all disciplines) between 2008 (8%) and 2013 (5%). Sig. increase over time for Indian Health Sciences journals between 2005 (54%) and 2015 (71%). | Higher UP of Spanish and Latin American Social Sciences journals (15%) in 2015 compared to Croatian Social Sciences journals in 2013 (2%). | No sig. difference in 2005 between MEDLINE indexed Indian Health Sciences journals and non-indexed journals (summary UP 58%). Sig. difference in 2008 between MEDLINE indexed Indian Health Sciences journals (60%) and non-indexed journals (35%). (possible influence of 26 journals that were not indexed in MEDLINE in 2005, but were indexed in 2008). | | Higher UP of Latin America and Caribbean Health Sciences journals in 2012 (61%) compared to Spanish and Latin American Social Sciences journals in 2015 (15%). | | No sig. difference in 1997 between South Korean Health Sciences Medical Association member journals and quasi-member journals (summary UP of 2%). Sig. difference in 2005 between ICMJE endorsing Indian Health Sciences journals (75%) and non-endorsing journals (49%). |

*The six disciplines we used in the study are: Arts & Humanities, Health, Life, Physical, Social, and Multidisciplinary Sciences. The specialties are sub-disciplines found within those areas (e.g., acoustics, botany, history, medicine, etc.).

Abbreviations and acronyms: Abridged Index Medicus (AIM), Chinese Medical Association Publishing House (CMPAH), Consolidated Standards of Reporting Trials (CONSORT), Directory of Open Access Journals (DOAJ), International Committee of Medical Journal Editors (ICMJE), Impact Factor (IF), Index Medicus (IM), Journal Citation Reports (JCR), U.S. National Library of Medicine bibliographic database (MEDLINE), PubMed Central (PMC), World Association of Medical Editors (WAME).

journals of different disciplines (Table 3 and Supplementary Information).

**Additional analyses**. Individual studies explored the association of seven additional factors with addressing of research integrity topics in ItAs, but again only for Health Sciences journals. The explored factors were language, publishers, endorsement of International Committee of Medical Journal Editors (ICMJE), endorsement of Consolidated Standards of Reporting Trials (CONSORT), membership in World Association of Medical Editors (WAME), in South Korean Medical Association (SHMA), or Chinese Medical Association Publishing House (CMPAH).

Except for language, for which one study in 2012 reported Iranian journals publishing in English covering authorship and conflicts of interest more often than journals publishing in Farsi[112], all other explored factors were found to be associated with some topics, while not with others (e.g., more ICMJE endorsing journals addressed ethics approval and URM, but not authorship or conflicts of interest when compared to non-endorsing journals).

Similarly, studies that reported on associations with impact factor values (but without providing data that could be used in

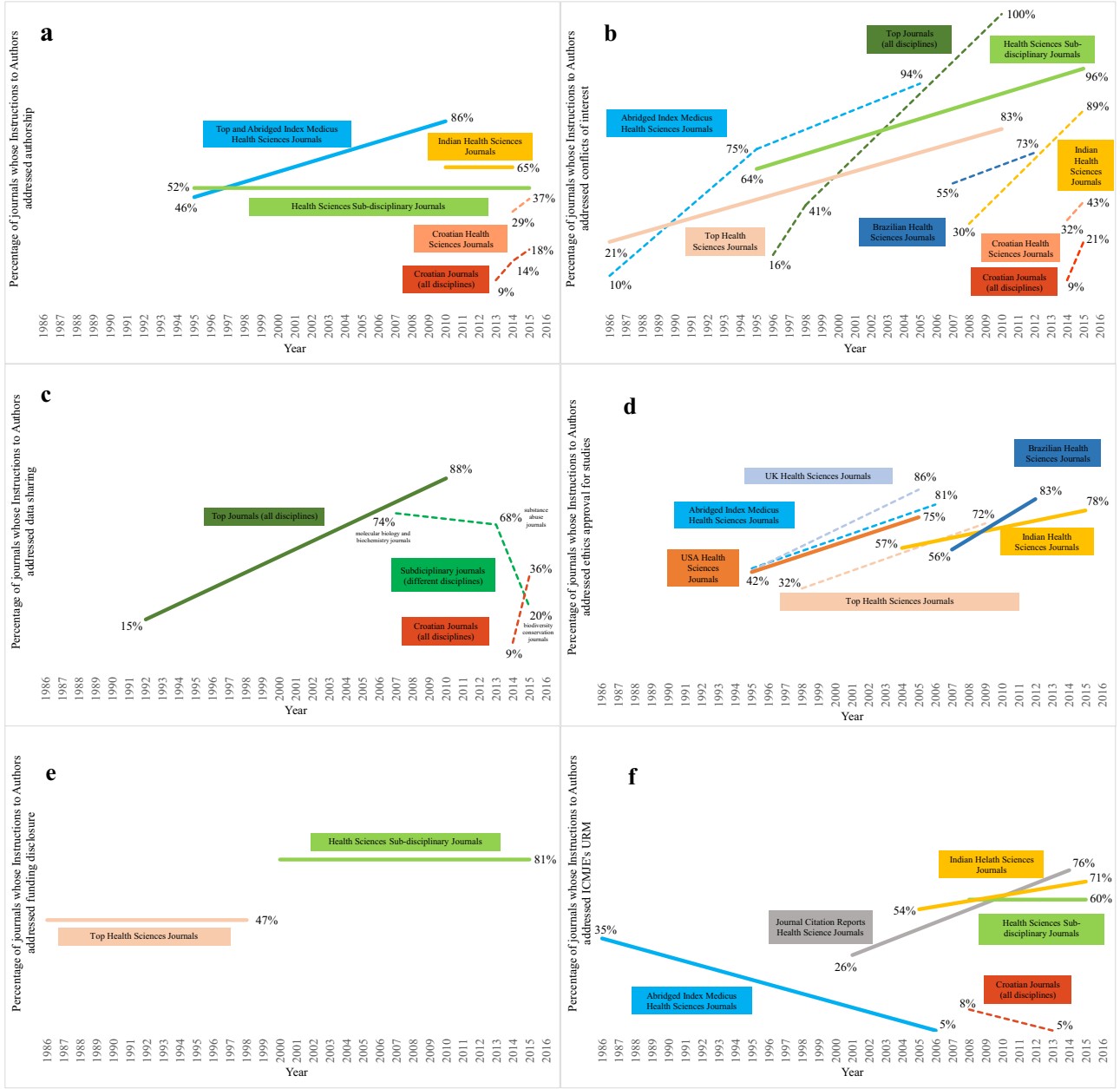

**Fig. 3 Changes over time in addressing publication ethics in journals' Instructions to Authors.** For ease of comparison all panels are on the same scale, with vertical axes based on a logit scale and percentages added as reference points. Panels represent changes over time for authorship (**a**), conflicts of interest (**b**), data sharing (**c**), ethics approval (**d**), funding (**e**), and International Committee of Medical Journal Editors Uniform Requirements for Manuscripts (**f**). Full lines represent trends obtained through regression models, while dash lines represent percentages reported in up to three studies.

meta-analyses) also reported conflicting results (Table 3 and Supplementary Information).

## Discussion

Our systematic review identified 153 studies that analysed journals' Instruction to Authors (ItAs) and 12 main objectives listed for conducting these studies, of which the most common were to determine whether journals had embraced particular policies or expert recommendations (e.g., reporting of ethics approval), and if papers published in those journals adhere to the journals' requirements. Such interest in journals' ItAs might reflect the attitudes and expectations of researchers that journals and editors should promote best scholarly practices and ensure the highest quality of papers they publish[175,176]. We also found an increase in the number

of studies analysing ItAs after 2002, which might reflect the switch to online publishing and the ease of obtaining digital instead of printed versions of ItAs and published papers, as well as the rise in numbers of meta-research studies over the last two decades[177].

Although there are indications, both in our study, and in the recently published scoping review[178], that recommendations or requirements stated in ItAs are associated with better study reporting, more studies are needed to identify the best methods for ensuring authors' compliance with ItA's, as well as for conducting editors' and reviewers' checks of that compliance. Future studies could also investigate how often and how well ItAs are read by the authors, and the effect ItAs might have on raising awareness of the topics they address.

Our series of meta-analyses on six research integrity topics (authorship, conflicts of interest, data sharing, ethics approval,

funding disclosure, and URM) found six factors that were associated with the addressing of those topics in journals' ItAs: time, country, database indexation, impact factor, discipline, and sub-discipline.

The overall increase in the number of journals addressing these topics over the last 30 years may be a results of several factors that include: the improvement of scholarly methods, the progress in teaching of those methods and standards of reporting[179,180], the rise in (inter)national regulations, especially regulations on obtaining ethics approval for studies[181], as well as increased attention to research integrity. The fact that most studies analysed ItAs of Health Sciences journals, and that Health Sciences journals covered these topics more frequently than journals of other disciplines, is most likely due to the strict regulations of experimentation on humans and animals, as well as increasing calls in Health Sciences for studies on editorial processes, peer review, research integrity, research waste and replication studies[18,175,182–184]. This could also indicate that Health Sciences journals might be leading the way in reporting practices for journals in other disciplines.

However, while the above findings paint an overall positive picture of the changes in ItAs over the last 30 years for these six research integrity topics, we found many exceptions to those trends. For example, ~52% of Health Sciences sub-disciplinary journals addressed authorship between 1995 and 2015; and there were also large differences in addressing of research integrity topics between countries (e.g., 14% of Croatian journals addressed authorship in 2014 vs. 70% of Indian journals). This indicates that many journals still lag behind recommending, requiring or implementing best reporting practices, which was also confirmed in our recent cross-sectional analysis of 19 transparency in reporting and research integrity topics across disciplines[12].

Furthermore, while we identified 153 studies that analysed the content of ItAs, the fact that most analysed only specific (sub) samples of journals and looked at addressing of only one or two topics within them, highlights the need for a comprehensive database that would allow authors or other stakeholders to compare journals based on their ItA requirements or recommendations (i.e., akin to SHERPA/RoMEO for listing of journal open-access and self-archiving policies[185], the Platform for Responsible Editorial Policies[186] or TOP Factor indicators[187]). Such a database could also include indicators of adherence of publications with the journal's requirements. Thus, it could function like Trials Tracker[188] for monitoring the compliance with EU or FDA regulations on timely posting clinical trial results. It could also enable mapping of changes in ItAs and journals' policies over time, while also providing a quality indicator or reputation safeguard for the journals.

Note that in the meta-analyses we conducted, we found associations for all six factors that we had data for, and that our narrative review showed indications of associations with an additional nine factors. And yet, many other factors potentially associated with the ItAs' contents, like the influence of specific editors or (large) publishers, changes made when a journal reaches a certain level of prestige[189], or (high profile) misconduct or legal cases, have not been explicitly explored in these studies (the exception being one study which showed differences between open-access publishing houses, professional organisation publishers, and other publishers for addressing of conflict of interest, but not for URM)[90].

Finally, both the meta-analyses we conducted and the percentages reported in individual studies on topics we did not meta-analyse, show that different topics follow different patterns, i.e., one topic being addressed in ItAs does not mean another one will be addressed too, nor that its coverage follows the same time trend, and so we warn against generalisation of the patterns we found for the six research integrity topics to other topics.

The strength of our study lies in the fact that we used the systematic review methodology to gather all studies analysing contents of ItAs of more than one journal, rather than focusing on a specific topic(s) or outcomes. But it also has several limitations. First, following our interests and project feasibility, we chose to meta-analyse only topics related to research integrity. Even though these topics were also among the most researched in the studies that analysed ItAs, further research should explore time trends and factors associated with addressing of other topics. Second, previous studies have shown that some enforced practices are not always listed in ItAs[190,191], while others, including studies listed in our narrative review, that listed practices are not always enforced[111,192], and finally, that some topics are reported in published papers even if not addressed in ItAs[105]. Afterall, ItAs are not meant to preclude authors for adhering to better reporting, and some authors are likely to go beyond (minimum) requirements imposed by the journals. Third, the association of countries, language and disciplines on reporting of research integrity topics have been demonstrated on a very small number of studies (1–5), of which the strongest indications come from two studies by the same author who looked at the ItAs of Croatian journals[143,144]. So further research into these associations is warranted. Finally, we have summarised information on addressing of topics in ItAs in a binary way (whether or not they were addressed), not on how each individual topic was addressed (e.g., the fact that authorship was addressed in ItAs, does not mean that all journals had the same requirements for authorship, nor that they addressed the number, order of authors, or the practice of shared authorship).

In conclusion, while our findings provide evidence that addressing of these six research integrity topics in journals' ItAs had increased over time, they also showed that many (sub)discipline and regional journals still lag behind in providing such guidance to authors. If publishers, editors and journals want to increase and safeguard the quality and transparency of reporting, they could benefit from updating and implementing policies that reflect and strengthen the integrity of research.

## Methods

We adhered to the Preferred Reporting Items for Systematic Reviews and Meta-Analyses (PRISMA) guidelines[193].

**Protocol and registration**. We could not preregister the study in PROSPERO as it did not include any health-related outcome; however, our projects' data repository site contains information on the conception of the study, as well as all the data and notes associated with it[174].

**Eligibility criteria**. We searched and included all studies that analysed ItAs of more than one journal, irrespective of the topic(s) ItAs were analysed for.

**Information sources**. We conducted the search on 1 May 2017 in three databases: MEDLINE (through Ovid interface), Scopus, and Web of Science (WoS) with no language or time restrictions. We also searched Google Scholar with the query -allintitle: instructions authors) -, and references of all included studies.

**Search**. The full search strategy for all three databases is available on our project's data repository site[174].

**Study selection**. We exported the search results of the three databases into Rayyan software[194], where manual deduplication was done by MM. Abstracts were assessed independently by MM and AJ to remove irrelevant studies. Disagreed upon studies were obtained in full (n = 25). Additional publications were identified through Google Scholar, through searching of references of selected studies, or through authors' awareness of published studies. Full texts of publications were checked by both assessors to confirm the eligibility criteria, and extract topics that were analysed in ItAs and the percentages of journals addressing those topics calculated based on all journals analysed in those studies.

**Data collection process and data items**. For each included study, MM extracted the following data in Excel: (1) number of journals whose ItAs were analysed within a study, (2) sampling method for choosing these journals, (3) discipline to which the analysed journals belonged to (reported disciplines were reclassified to fit the following categories: Arts & Humanities, Health Sciences, Life Sciences, Physical Sciences, Social Sciences, and Multidisciplinary Sciences), (4) sub-discipline to which the journals belonged to (as specified in the respective studies, e.g., dental medicine for Health Sciences), (5) countries or territories in which the journals were published, (6) year when the journals' ItAs were accessed/analysed, (7) topic(s) that were analysed, (8) number and percentage of journals addressing a topic (out of the total number of journals whose ItAs were analysed in a study), (9) method(s) of analysing the ItAs (e.g., one or more researchers reading the ItAs), (10) factors explored for possible association with addressing a particular topic (e.g., journal's impact factor, indexed database, or publisher), and (11) objectives or hypotheses listed as reasons for conducting the study. The (names of) databases in which the journals were indexed were extracted as reported in original studies, we did not assign database indexation to studies in which they were not reported as only a third of studies reported a full list of journals they analysed ($n = 53$, 35%, see below). For all included studies AJ then checked if the data extraction was done correctly.

Additionally, the following variables were also extracted, but were not included in the results synthesis: (1) if the authors also surveyed editors about questions relating to the submission process, (2) if the studies included a detailed list of journals whose ItAs were analysed.

The data extraction process revealed that >100 different topics were analysed across the studies, and so we grouped them into major topic variable, while we also kept a record from which studies included other (sub)topics for which we did not extract the data (all recorded sub-topics are available at our project's data repository site)[174]. To clarify, we list here two examples of sub-topics: (1) for the topic Reporting Guidelines, the sub-topics were: mentioning of different specific reporting guidelines; (2) for the topic Case Reports (mentioning of publishing case report studies in the journal), the sub-topics were: maximum word count allowed or requiring a structured abstract.

Finally, on 4 December 2020, we extracted the number of journal articles in PubMed, WoS, Scopus and Crossref. Searches and extracted numbers are available on our project's data repository site[174].

**Synthesis of results and additional analysis**. No meta-analyses were predefined at the study conception stage, as the number of topics analysed and the factors explored as determinants of journals addressing a topic could only be assessed after conducting the systematic review.

To study the time trend in the number of studies analysing ItAs and the number of journal publications in PubMed, WoS, Scopus and Crossref, we used a spline regression model that was fitted to the data using nonlinear regression within the SPSS v24 software (IBM, Chicago, IL, USA). Comparisons of number of published studies analysing ItAs published ≤2002 vs. >2002, with number of articles published in PubMed, WoS, Scopus, or Crossref in the same period were conducted with a series of chi-squared tests.

Meta-analyses of percentages of journals addressing a particular research integrity topic in ItAs were performed using Comprehensive Meta-analysis Software (CMA) version 3 (Biostat Inc., Englewood, USA), which pools percentages using the logit transformation method. The percentages with their 95% confidence intervals (CI) were calculated within CMA software from the number of journals addressing a topic out of the total number of journals analysed in a particular study. If percentages reported in a study were 0% or 100%, CMA introduced a continuity correction to avoid including studies with standard errors of zero.

Random-effects models were used to estimate summary percentages. However, because estimation of random-effects models with few studies have been shown to be unreliable[195], if fewer than five studies were included in a meta-analysis we applied a fixed-effect model. Statistical heterogeneity of studies was estimated with both Cochran's Q test, and Higgins's $I^2$ test statistic. In case of considerable heterogeneity, we did not pool the data, as listing a summary percentage could be misleading. In those cases, to explain the heterogeneity, we conducted mixed-effects subgroup analyses (for categorical factors such as countries, disciplines, or impact factor categories reported in the studies) or random-effects meta-regressions based on the DerSimonian and Laird method, which doesn't assume that effect sizes are normally distributed (for the factors time, impact factor values, discipline, or indexed database). We also searched for sources of heterogeneity if percentages were dispersed throughout the 0 to 100% interval, but due to high uncertainty 95% CIs largely overlapped). If the number of studies was too small to perform meta-regression and a factor was numerical (i.e., for $n < 4$ when the factor time was assessed), we decided that a factor significantly affects the percentages if we could show: (a) consistency in direction of change in percentages with growing values of the factor, and (b) significant differences between percentages assigned to neighbouring values of a factor. Differences in percentages reported in two studies, together with associated 95% CIs for the difference, as well as $p$-values for statistical test of differences, were estimated by Exploratory Software for Confidence Intervals (ECSI) software[196]. Pseudo-$R^2$ index was used to quantify the proportion of variance explained by a factor.

A $p$-value of <0.05 was considered to indicate statistical significance. However, when analyses were underpowered (e.g., for comparison of two estimates with one being made on a sample of five journals), we also stated if a result was significant at the 0.1 level. All the tests were two-sided.

**Risk of bias**. We are aware of no tools that measure the risk of bias of studies whose units of analysis are ItAs. As the methodology of these studies involves selecting journals for analysis, obtaining ItAs from printed volumes or downloading them from journal's websites, and extracting data on topics addressed in ItAs by reading them (by one or more researchers) or using qualitative text analysis software, we included all eligible studies in synthesis of results and meta-analyses. We, however, did provide notes, in both Table 3 and the Supplementary Section 2, about methodological factors, which could have an effect on the validity and reliability of reported estimates. In regards to selection bias, none of the studies used probability sampling that would cover all disciplines and a wide range of journal (citation) influences or indexing databases. Rather, most studies analysed ItAs of Health Sciences journals ($n = 116$, 76%), and sampled only journals that were indexed in the Journal Citation Reports database ($n = 55$, 47%). In regards to reporting bias, studies often omitted explaining their methods of analysing ItAs ($n = 94$, 61%), or listing the year of ItAs that were analysed ($n = 69$, 45%, Table 1).

## Data availability

The data generated in this study have been deposited in the Mendeley repository with the identifier: https://data.mendeley.com/datasets/53cskwwpdn/5 [174].

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

## Acknowledgements

We would like to thank Ana Utrobičić for her help with developing the search strategy. We would like to thank Anna Tordai for help with data extraction and translation of articles in Hungarian and French, Sjors de Heuvel for an article in Japanese, Lionel Dias for articles in Spanish and Portuguese, Yong Hu for an article in Chinese, and Natalia Lee for an article in Korean. Finally, we would like to thank Bianca Kramer for her help in search strategies to extract number of publications in Crossref, PubMed, Scopus, and Web of Science. This study was a part of the Elsevier funded project Fostering Transparent and Responsible Conduct of Research: What can Journals do? The work of AJ was supported by the Croatian National Science Foundation (HRZZ IP-2018-01-4729).

## Author contributions

M.M.: conceptualisation, data curation, formal analysis, investigation, methodology, visualisation, writing—original draft preparation, writing—review & editing., A.J.: conceptualization, data curation, formal analysis, investigation, methodology, visualisation, writing—original draft preparation, writing—review & editing, I.J.J.A.: conceptualization, methodology, funding acquisition, writing—review & editing, L.B.: conceptualization, methodology, funding acquisition, writing—review & editing, G.t.R.: conceptualization, methodology, project administration, supervision, writing—review & editing.

## Competing interests

I.J.J.A. is Senior Vice-President of Research Integrity for Elsevier. M.M. is a Co-Editor-in-Chief of BMC's Research Integrity and Peer Review journal. G.t.R. was an author of one study included in this review[49], however the inclusion of the study in the review, as well as data extraction and analysis were conducted by M.M. and A.J. The remaining authors declare no competing interests.
