## [Peer Review File · Nature Communications]

Reviewer #1 (Remarks to the Author):

Thank you for writing and submitting this manuscript. Comments are below.

Notes on the major claims of the paper

Essentially you reported that instructions to authors varied over time and some aspects evolved over time. Journals with higher impact factors were more robust about instructions to authors than journals with lower impact factors in their disciplines. While these are expected and present nothing new, it was not clear to me reading your research why it would need a systematic review or "meta-analysis" in the first place. While the facts you have reported are intuitive, you did not discuss if there were differences in the extent or variations that needed to be resolved using a systematic review.

Essentially, it was not clear to me while reading the introduction why a meta-analysis as opposed to a simple narrative review was warranted or what additional benefit would a "meta-analysis" add to what is known already. Unless there is a reason to resolve extant disagreements, a systematic review is not warranted in a research question where you essentially asked "what do we know of the instructions to authors of journals"

Innovativeness and intellectual appeal to others in the community and the wider field

The research itself has no theoretical or applied merit that can change any practice. There is no reason to believe that beyond staking a claim that this is in pursuit of curiosity, becoming more explicit in instructions to authors would influence a journal's rankings in terms of impacts or citation metrics, nor was this explored in the paper either. Essentially, you restated and summarised the conclusions of the other papers.

Potential to influence thinking in the field

Practically nothing as the results are intuitive but a numerical metric is not warranted nor likely to make a difference.

Notes on reproducibility of the work, given the level of detail provided.

It was not clear what was your hypothesis, and what were the exact search terms in addition to Google Scholar "allintitle:instructions authors" possibly for all years? When I conducted the same search, it retrieved only seven resources, and it was not clear on the basis of what would you select the articles for your review.

From your detailed description, it was not clear why you selected a "random sample". This goes against any attempt at comprehensive meta analysis or systematic review or synthesis of all available information on a given topic. Besides, unlike humans or animals, journals are not random entities and each journal is unique in ways that are pertinent to the extent to which they might issue instructions to the authors. It'd be one thing to conduct an eclectic review of journals for a particular field, or a set of fields, but that would not need a "systematic review" or "meta-analysis". Hence, in your revision, can you kindly explain in details as to why it'd be pertinent to treat all journals as uniform.

Lastly, what you conducted cannot be considered a "meta-analysis" in the conventional sense.

Reviewer #2 (Remarks to the Author):

This article reports on an interesting piece of research, namely to conduct a review of reviews examining instructions to authors.

I think the rationale for the research needs to be stated more vehemently as does the particular

method and what this is adding to the field, why this is original and beneficial. The rationale and the following sentences stating the particular benefits the study will deliver are a little muted.

In the results section, some of the language needs to be more specific to aid readers' comprehension. I have added some suggestions and ideas for the authors' consideration.

The methodology and methods are reported clearly, with good use of tables to present data. The search strategy does not appear to be on the Mendeley site, so either needs adding or needs renaming to aid discovery.

The data highlighting the lack of adherence to ItA are revealing and begs questions about the peer-review and scholarly publishing process in ensuring adherence to these criteria. The work will be of interest to all those in the research system, particularly those interested in 'research on research'.

I have added some additional comments in the text that the authors may find helpful in fine tuning the article.

Reviewer #3 (Remarks to the Author):

This is an interesting and relevant study of the evolution of ItAs across 30 years and 6 primary research integrity topics, in addition to a meta-analysis of six factors that emerged as crucial to explaining why the study findings were so disparate. Dr. Malicki and this team have a reputation for pushing for more transparent reporting of studies, data sharing, and other practices that create more responsible and reproducible science. It is heartening that they found a trend that more research integrity topics are being addressed by ItAs over time. This is a useful baseline report on ItAs that could be repeated periodically to help gauge progress in research integrity. This study will be of interest to the research community. As I read it, I wished that there was more comparison across disciplines, but I now understand that this is a companion piece to another study that does more analysis.

General: The work is generally well-written and understandable, despite the immense amount of data to be digested. Readers will need to spend some time with Table 3 to follow the write-up. The study is well framed in the literature and previous research on this and adjacent topics.

One small irritant for me was the overuse of the phrase "on addressing of xxx" which appears throughout the manuscript. It is awkward to use the gerundive noun as a shorthand in ways that impact the readability of the study. The authors need to come up with another verbal formula, vary the use of this phrase, introduce it as a described term at the beginning, or simply use more words to describe this each time – or some combination of all of the above.

Title: Perhaps consider adding the dates, since the search strategy is a little dated? Add 1986-2017 (2016?).

Table 2. Does the growth in the number of publications analyzing ItAs mirror the general growth in publishing? This should be noted.

Abstract: I would have liked to see a concluding line that indicated some of the key findings of the paper or at least an acknowledgement that this paper mostly establishes a baseline of findings. The construction "higher addressing" does not make sense.

The description of factor 1 (time) seems like it should say "over time" - "mostly increased over time"

Introduction: The paper needs minor editorial review for a few misplaced punctuation and infelicitous

word choices ("depict methods" – maybe "describe methods").

RESULTS:

Series of Meta-Analyses:

Instead of "for addressing of six" try just "to address six". Should "time" be "variation over time"?

Country:

The introductory paragraph is confusing. Percentages [of ItAs] addressing? Needs editing.

Journal Indexation:

I wish that a Methods section had more clearly explained the different groupings that the authors use: AIM, JCR, DOAJ – I'm still not clear on when, how and why they were using different published lists. I am also not clear what lists the authors used to categorize journals outside of the Health Sciences.

Additional Analyses:

I find it odd that ICMJE-endorsing journals did not address authorship and conflicts of interest as much as the other groups. There were times I wished for more discussion than description with this study.

DISCUSSION:

While the heterogeneity of the papers is well-noted, I think that there were more interesting themes that could have been extracted from the rich data. For example, why did 61% of studies not specify their analytic method?

Minor errors throughout – I assume will be corrected in the editorial process:

First paragraph: lost) in line 4. "Conducting these studies" (no "of"). Increase in the number of studies" [not "of"]. Maybe add the word "gradual" to the claim about 2002? When were ItAs first online? Well before 2002, I would think, so the "switch to online publishing" as a cause might be more fully explored.

Second paragraph: "Although" not "Though". Drop "the" between "that better reporting of".

I think that you should add a sentence recognizing the role of peer reviewers and editors in helping to ensure authors' compliance with ItAs as well.

Paragraph 4: Confusing. "Probably corresponds to the improvement" "increased attention to research integrity."

Why did no Health Sciences papers discuss URM? One concern is the vast representation of Health Sciences journals in the final set of studies analyzed. Were there more comparisons that could be made within this subset of studies? Perhaps this is a separate paper that the team intends to publish; this would be of great interest.

Paragraph 6 – the suggestion of a database to compare publications' ItAs is a wonderful idea. I hope that the team can help implement such a project. "for timely posting of clinical trial results".

Paragraph 9 – While briefly mentioned, the concept that some authors will always be more rigorous than required by ItAs should be strengthened. ItAs are minimum standards and the most rigorous papers will often exceed the requirements stated, incorporating other standards and emerging best practices. ItAs should never preclude authors for adhering to better reporting.

METHODS

Information Sources and Search

The authors state that the full search strategies for all three databases are available on their project repository. I have scoured that site and downloaded the full data files and cannot find the search strategies. While I am not inclined to doubt the validity of the searches – librarian Ana Utrobicic is acknowledged – I find it troubling that the search strategies for any systematic review are so difficult to find and verify since the search strategy is the foundation of any systematic review. The text refers

to searches done on 1 May 2017, which is over three year ago. While analyzing the volume of data for this project undoubtedly took a significant amount of time, I am concerned that the search was not updated before the manuscript was submitted. Considering that there were only 812 studies to screen originally, an update would not have taken long. Even a brief literature search shows that additional papers on this topic have been published, although I don't know if they would have been included in the analysis because the authors don't clearly state their inclusion or exclusion criteria in either the text or PRISMA flow chart. Also, they do not appear to have subjected the included studies to any appraisal of quality. Overall, the systematic review aspects of the study could have been more successfully reported.

Conclusion:

Is there a role for a "lessons learned" section or advice for others who might want to replicate this study?

With their commitment to standards and reporting, I would hope that this team would also conform to the emerging standards for reporting searches for systematic reviews – PRISMA-S (<https://osf.io/ygn9w/>). A baseline study such as this one will be difficult to reproduce if the searches are not documented and available. Including the search strategy of at least one of the searches in the appendix published with the article would be preferable to using an ephemeral data repository.

I am not trained to evaluate the statistical analysis of this paper.

I withhold judgement on the overall paper until I can review the search strategies. But I am likely to recommend the publication of this manuscript with minor revisions.

Holly Grossetta Nardini

Dear Editor Nathalie Le Bot,

thank you and your reviewers for the constructive suggestions to improve our manuscript. We believe we have addressed all suggestions appropriately, and we present them below in a point-by-point manner.

Editors comments:

#1 For us to consider the manuscript further, it will be essential to follow more closely what is expected from the approach of a systematic review. For example, reviewer #1 outlined in comments to the editors that it would have been more appropriate to first discuss that wide variability exists and then explore the variability and summarising the components that make up this variability.

Reply: We thank the editor for this comment, and while it might be more common for health-outcomes research to first discuss variability in studies and then conduct a systematic review and meta-analysis to address the variability, this was not the case in our research. Our study was conducted for two main reasons: 1) to identify and summarize all research on journals' instructions to authors in an unbiased and replicable way (this exploratory, unbiased approach, is one of the main reasons for conducting systematic reviews and research synthesis - see Chalmers, Hedges and Cooper: Brief History of Research Synthesis, and 2) use that knowledge to inform our own study on journals' instructions to authors. And these two reasons have been preregistered on our project website. Only after conducting the systematic review, did we see that not only large variability existed, but that 15 factors have been explored as reasons behind this variability. And it was at this point that we decided to conduct meta-analyses to try to resolve many conflicting results of the individual studies. This is now clearly stated in the revised paper. As you may remember, we first submitted the systematic review as a stand-alone paper to your journal, and then on yours and the editorial boards' suggestion we merged it with the meta-analyses which were also initially submitted as a sister paper to the sys. review.

#2 Reviewer #3, a second expert on systematic reviews, also notes that it would have been interesting to go beyond establishing the baseline by exploring further the rich dataset. We strongly encourage you to include such analysis in a revised manuscript.

Reply: We thank the editor for this comment, but we would like to point out that our database has been fully explored for all factors reported in individual studies for the 6 research integrity topics we cover. A plethora of (sub-)analyses are presented in the appendix (pages 8 to 32). We therefore feel that we went far beyond establishing a baseline. We also think that reviewer #3's comments referred to their wish to know more about differences in ItAs between disciplines and results for URM, which were fully explored in the appendix and are mentioned in our results. Perhaps, reviewer #3 did not check our appendix fully (see our responses to the reviewer #3 below). As most studies analysed ItAs of Health Science journals, even with 153 studies included in our systematic review, evidence on differences between disciplines is largely lacking. To explore these differences, we conducted an additional study, and that study's results are mentioned in our discussion (study link: <https://doi.org/10.1371/journal.pone.0222157>).

#3 In addition, all reviewers outline that the current manuscript lacks information on the approaches taken to gather the information and these concerns would have to be addressed in full. We also note that the revised manuscript needs to follow PRISMA guidelines for reporting.

Reply: The information on the searches and all elements of PRISMA are now included.

#4 In addition to the above, you must comply with the following editorial requests; we will not be able to proceed with your revised manuscript otherwise. Please also see the Nature Communications formatting instructions, which you may find useful while preparing your revised manuscript.

Reply: We have followed the formatting instructions in the revised manuscript. The only exception was to keep the formatting of Table 2 due to its complexity and size.

#5 Please complete or update the following checklist(s) to verify compliance with our research ethics and data reporting standards. Address all points on the checklist, revising your manuscript in response to the points if needed. The form(s) must be downloaded and completed in Adobe Reader rather than opened in a web browser. Each form must be uploaded as a Related Manuscript file at the time of resubmission. Editorial policy checklist:

<https://www.nature.com/documents/nr-editorial-policy-checklist.pdf>

Reporting summary:

Reply: Both the editorial policy checklist, and the reporting summary checklist have been completed and uploaded with the revised manuscript.

#6 All Nature Communications manuscripts must include a “Data Availability” section after the Methods section but before the References. If any of the data can only be shared on request or are subject to restrictions, please specify the reasons and explain how, when, and by whom the data can be accessed. For more information on this policy and a list of examples, see:

<https://www.nature.com/documents/nr-data-availability-statements-data-citations.pdf>

We strongly encourage you to deposit all new data associated with the paper in a persistent repository where they can be freely and enduringly accessed. We recommend submitting the data to discipline-specific and community-recognized repositories; a list of repositories is provided here: <http://www.nature.com/sdata/policies/repositories> Refer to our data policies here: <https://www.nature.com/nature-research/editorial-policies/reporting-standards#availability-of-data>

* To maximise the reproducibility of research data, we strongly encourage you to provide a file containing the raw data underlying the following types of display items:

- Any reported means/averages in box plots, bar charts, and tables
- Dot plots/scatter plots, especially when there are overlapping points
- Line graphs

The data should be provided in a single Excel file with data for each figure/table in a separate sheet, or in multiple labelled files within a zipped folder. Name this file or folder ‘Source Data’, and include a brief description in your cover letter. The “Data Availability” section should also include the statement “Source data are provided with this paper.”

To learn more about our motivation behind this policy, please see: <https://www.nature.com/articles/s41467-018-06012-8>

Reply: The data availability statement is included in the revised manuscript and data is provided in a single excel file deposited on our project’s website.

#7 ORCID Nature Communications is committed to improving transparency in authorship. As part of our efforts in this direction, we are now requesting that all authors identified as

'corresponding author' create and link their Open Researcher and Contributor Identifier (ORCID) with their account on the Manuscript Tracking System prior to acceptance. ORCID helps the scientific community achieve unambiguous attribution of all scholarly contributions. You can create and link your ORCID from the home page of the Manuscript Tracking System by clicking on 'Modify my Springer Nature account' and following these instructions. Please also inform all co-authors that they can add their ORCIDs to their accounts and that they must do so prior to acceptance. For more information please visit <http://www.springernature.com/orcid>

Reply: ORCID iDs are listed for all authors.

#8 Please use the link below to submit the following items as separate documents:

- Revised manuscript
- Any supplementary files
- Point-by-point response to the reviewers' comments, reproduced verbatim
- Cover letter to the editor
- Any completed checklist(s)

Reply: All of the above was uploaded with the revised manuscript.

REVIEWER COMMENTS

Reviewer #1 (Remarks to the Author):

Thank you for writing and submitting this manuscript. Comments are below.

Notes on the major claims of the paper

Reviewer #1: Essentially you reported that instructions to authors varied over time and some aspects evolved over time. Journals with higher impact factors were more robust about instructions to authors than journals with lower impact factors in their disciplines. While these are expected and present nothing new, it was not clear to me reading your research why it would need a systematic review or "meta-analysis" in the first place. While the facts you have reported are intuitive, you did not discuss if there were differences in the extent or variations that needed to be resolved using a systematic review. Essentially, it was not clear to me while reading the introduction why a meta-analysis as opposed to a simple narrative review was warranted or what additional benefit would a "meta-analysis" add to what is known already. Unless there is a reason to resolve extant disagreements, a systematic review is not warranted in a research question where you essentially asked "what do we know of the instructions to authors of journals"

Reply: We respectfully disagree with the reviewer that a systematic review is not warranted unless it is meant to resolve disagreements. Good systematic reviews are a form of a unbiased, methodologically rigorous research synthesis and are preferred over narrative reviews (see Chalmers, Hedges and Cooper: Brief History of Research Synthesis, and Munn et al. What kind of systematic review should I conduct?). Whether a systematic review is followed by a meta-analysis depends on several factors, one of which can be the desire to resolve controversy with conflicting results of prior studies (Anello & Fleiss; Exploratory or analytic meta-analysis: should we distinguish between them?). In our case, many controversies did exist surrounding addressing of topics in instructions to authors – especially regarding associations with impact factor and changes over time, but these became known to us, only after conducting the

systematic review. Furthermore, although the reviewer finds these results intuitive, we do not see how a narrative synthesis might have answered these questions. Note also that even the results of our meta-analyses provide only limited evidence for the role of time and impact factor. We show that instructions have neither improved steadily for all disciplines or countries over time, nor that the journals with higher impact factors are unequivocally more detailed in their instructions. We only confirmed differences between the top and the lowest ranked *Health sub-disciplinary* journals. Finally, and importantly, our reasons for conducting the systematic review were preregistered on our project's website. We initially submitted the systematic review as a stand-alone paper to Nature Communications and the meta-analyses as a second paper, but the editors suggested we merge them together. We have clarified in the revised version, that the goal of the systematic review was to collect and summarize all research on instructions to authors, and the goal of meta-analyses to resolve disagreements as well as explore additional factors (e.g., methodological differences between studies) as potential reasons for those disagreements.

Innovativeness and intellectual appeal to others in the community and the wider field

Reviewer #1: The research itself has no theoretical or applied merit that can change any practice. There is no reason to believe that beyond staking a claim that this is in pursuit of curiosity, becoming more explicit in instructions to authors would influence a journal's rankings in terms of impacts or citation metrics, nor was this explored in the paper either. Essentially, you restated and summarised the conclusions of the other papers.

Reply: We thank the reviewer for their comment, but again respectfully disagree. Our research is a historical overview and synthesis of studies that analysed journals' Instructions to Authors (ItAs). Like all systematic reviews, it presents a summary of results of previous studies, while the meta-analyses provide additional insights by exploring potential reasons behind conflicting findings in individual studies. Our study has shown that some of the individual studies aimed to provide recommendations or criticize journals' editors and the content of ItAs, while others aimed to explore the influence of ItAs on reporting of studies. We however believe our results will indeed change practice by encouraging editors to update their ItAs (I personally have already updated instructions of the journal where I am an editor), and that it will lead to creation of a database capturing ItAs of all journals in one place (akin to recent database of capturing all editors <https://openeditors.ooir.org/>). And finally, we hope that our paper will help researchers conducting these types of studies to avoid their predecessors' limitations, and even encourage design of studies that will answer the outstanding questions which we mention in the discussion.

Potential to influence thinking in the field

Reviewer #1: Practically nothing as the results are intuitive but a numerical metric is not warranted nor likely to make a difference.

Reply: We thank the reviewer for their honest opinion. However, we have pointed out above that the results the reviewer finds intuitive have not been unequivocally confirmed even by our meta-analyses. Many countries and disciplines have shown no improvements in the ItAs content for decades, and for many topics' associations with impact factor of journals were not found. That said, as we stated above, we hope our results will prompt more rigorous studies of journals ItAs, their updates, and a creation of a database similar to Platform for Responsible Editorial Policies (PREP) that would allow direct comparison of ItAs contents for all journals in existence. Finally, we would like to point the reviewer to the opinions of the reviewers #2 and #3 about the potential influence our study will have on the field.

Notes on reproducibility of the work, given the level of detail provided.

Reviewer #1: It was not clear what was your hypothesis, and what were the exact search terms in addition to Google Scholar "allintitle:instructions authors" possibly for all years? When I conducted the same search, it retrieved only seven resources, and it was not clear on the basis of what would you select the articles for your review.

Reply: I apologize for the lack of search strategies on our project website, I believed I had uploaded it, but I failed to double-check. This has been corrected and full search strategies for all databases are now available on <https://data.mendeley.com/datasets/53cskwwpdn/5>. As for the hypothesis, as we already stated above, we did not have one at the start of this project – our aim was to find and summarize all studies that analysed ItAs. We however now explain that the reason for meta-analyses and meta-regressions was to resolve discrepancies and to explore the heterogeneity we observed between the studies.

Reviewer #1: From your detailed description, it was not clear why you selected a "random sample". This goes against any attempt at comprehensive meta-analysis or systematic review or synthesis of all available information on a given topic. Besides, unlike humans or animals, journals are not random entities and each journal is unique in ways that are pertinent to the extent to which they might issue instructions to the authors. It'd be one thing to conduct an eclectic review of journals for a particular field, or a set of fields, but that would not need a "systematic review" or "meta-analysis". Hence, in your revision, can you kindly explain in details as to why it'd be pertinent to treat all journals as uniform.

Reply: We believe that the reviewer has misunderstood something. We have not drawn any random samples. We have included all studies that analysed ItAs. Random sampling is only mentioned in our manuscript's appendix or in our results' tables, and it refers to the way *individual researchers* selected journals whose ItAs they analysed. The rationale the authors of those studies had was to select a representative sample of journals from a database or discipline, instead of only top or conveniently sampled number. The population of journals today has a particular size, and like any population – if one is making claims and describing the content of ItAs of all journals or journals from a specific discipline – a random sample is the best approach, second only to census.

Reviewer #2 (Remarks to the Author):

This article reports on an interesting piece of research, namely to conduct a review of reviews examining instructions to authors.

I think the rationale for the research needs to be stated more vehemently as does the particular method and what this is adding to the field, why this is original and beneficial. The rationale and the following sentences stating the particular benefits the study will deliver are a little muted.

Reply: We thank the reviewer for the comment and have emphasized the rationale and the findings in the revised manuscript.

In the results section, some of the language needs to be more specific to aid readers' comprehension. I have added some suggestions and ideas for the authors' consideration.

Reply: We thank the reviewer for the suggestions and have incorporated many language changes in the revised manuscript.

The methodology and methods are reported clearly, with good use of tables to present data. The search strategy does not appear to be on the Mendeley site, so either needs adding or needs renaming to aid discovery.

Reply: The search strategies have been added to our projects website (Mendeley), my apology for this omission during the initial review. We have also included the search strategy at the end of the rebuttal letter.

The data highlighting the lack of adherence to ItA are revealing and begs questions about the peer-review and scholarly publishing process in ensuring adherence to these criteria. The work will be of interest to all those in the research system, particularly those interested in ‘research on research’. I have added some additional comments in the text that the authors may find helpful in fine tuning the article.

Reply: We thank the reviewer for the comment and the suggestions, which we have incorporated in the revised version.

Reviewer #3 (Remarks to the Author):

This is an interesting and relevant study of the evolution of ItAs across 30 years and 6 primary research integrity topics, in addition to a meta-analysis of six factors that emerged as crucial to explaining why the study findings were so disparate. Dr. Malicki and this team have a reputation for pushing for more transparent reporting of studies, data sharing, and other practices that create more responsible and reproducible science. It is heartening that they found a trend that more research integrity topics are being addressed by ItAs over time. This is a useful baseline report on ItAs that could be repeated periodically to help gauge progress in research integrity. This study will be of interest to the research community. As I read it, I wished that there was more comparison across disciplines, but I now understand that this is a companion piece to another study that does more analysis.

Reply: We thank the reviewer for the kind words. This is indeed a companion study of our large project on ItAs and attitudes of researchers towards research integrity topics. We are however unable to provide more exploration of differences between disciplines based on this dataset as too few studies explored such differences. We reported all those that did in our results and appendix. This has also prompted us to conduct a study to answer this specific question – and if the reviewer is interested, they may see the results of that study here:

<https://doi.org/10.1371/journal.pone.0222157>. We also discuss that study and its results in the discussion section of the current manuscript.

General: The work is generally well-written and understandable, despite the immense amount of data to be digested. Readers will need to spend some time with Table 3 to follow the write-up. The study is well framed in the literature and previous research on this and adjacent topics.

Reply: We thank the reviewer for these kind words.

One small irritant for me was the overuse of the phrase “on addressing of xxx” which appears throughout the manuscript. It is awkward to use the gerundive noun as a shorthand in ways that impact the readability of the study. The authors need to come up with another verbal formula, vary the use of this phrase, introduce it as a described term at the beginning, or simply use more words to describe this each time – or some combination of all of the above.

Reply: We thank the reviewer for this suggestion. The manuscript has now been revised, and includes both the terms addressing the topics and covering the topics, and we have also removed unnecessary instances of *addressing* throughout the manuscript. We also made many additional stylistic and wording changes.

Title: Perhaps consider adding the dates, since the search strategy is a little dated? Add 1986-2017 (2016?).

Reply: We have added the dates and changed the title to be in line with the journals' formatting recommendations.

Table 2. Does the growth in the number of publications analyzing ItAs mirror the general growth in publishing? This should be noted.

Reply: We thank the review for this question. On 4 December 2020, we extracted information on all journal articles in PubMed, WoS, Scopus and Crossref and present that data alongside the data on ItA studies in the new revised Figure 2 (also shown below). Growth of studies that analysed ItAs was faster than that of that of publications in general (confirmed by chi-squared tests of number of studies ≤ 2002 vs > 2002 , with P values of < 0.0001). We have therefore stated in the results that the “*growth was faster than of article publications in that period, chi-squared tests, $P < 0.001$ for all comparisons*”. (Note for the editor: we also updated our methods to include these additional analyses, as well as our acknowledgment section to thank Bianca Kramer for the help with strategies for data for the revised Figure 2).

Figure 2. Growth of the number of publications analysing journals' Instructions to Authors (ItA), alongside the growth of journal articles on Crossref, PubMed, Scopus and Web of Science. Prediction lines were determined by optimal spline regression models.

Abstract: I would have liked to see a concluding line that indicated some of the key findings of the paper or at least an acknowledgement that this paper mostly establishes a baseline of findings.

Reply: The key findings have been stressed, and the fact that we provide a timeline of changes has now been added. Additionally, in line with Nature Communications instructions, a summary of our results is now also presented in the last paragraph of the introduction.

The construction “higher addressing” does not make sense.

Reply: We removed the expression and changed it throughout the manuscript, for example “*addressing was more common in journals with highest impact factor values*”.

The description of factor 1 (time) seems like it should say “over time” - “mostly increased over time”

Reply: Over time was added and corrected, thank you for noticing this.

Introduction: The paper needs minor editorial review for a few misplaced punctuation and infelicitous word choices (“depict methods” – maybe “describe methods”).

Reply: We have corrected misplaced punctuations and changed several expressions throughout the manuscript.

RESULTS:

Series of Meta-Analyses:

Instead of “for addressing of six” try just “to address six”. Should “time” be “variation over time”?

Reply: We thank the reviewer for the suggestions, these and many other stylistic changes have been incorporated in the revised manuscript.

Country:

The introductory paragraph is confusing. Percentages [of ItAs] addressing? Needs editing.

Reply: This paragraph was been rewritten.

Journal Indexation:

I wish that a Methods section had more clearly explained the different groupings that the authors use: AIM, JCR, DOAJ – I’m still not clear on when, how and why they were using different published lists. I am also not clear what lists the authors used to categorize journals outside of the Health Sciences.

Reply: We thank the reviewer for pointing out that this needs clarifying. We have now written in the methods that the groupings are as were reported in individual studies. We could not regroup the journals to the categories ourselves, as lists (and names) of analysed journals were only reported in 35% (n=53) of primary studies. Outside of Health Sciences, authors used, for example, Scopus journal classification, JCR journal classification, national database classifications, and similar (full list of classifications are available in our raw data file).

Additional Analyses:

I find it odd that ICMJE-endorsing journals did not address authorship and conflicts of interest as much as the other groups. There were times I wished for more discussion than description with this study.

Reply: We thank the review for the comment, but it is difficult to discuss each finding of individual studies within a limited space of one systematic review paper and meta-analysis paper, especially in light of 153 included studies. We focused most of our discussion on the six main factors identified in the meta-analysis and patterns identified in the results synthesis. In our appendix, which alone has 41 pages, the readers can find much more information and even our comments on this particular finding - that it might be a by-product of the small sample size of that study. In the discussion section of that study, authors themselves stated that formatting, authorship and COI have always been the most covered topics in journals ItAs, and that many journals defined authorship and COI differently than in ICMJE, so endorsement of ICMJE was more likely lead to change in how it was defined (e.g., adopting ICMJE COI disclosure form), rather than mentioning it anew. We would also like to point out that journals endorse ICMJE for specific topics and that such endorsement(s) often did not mean they aligned with all recommendations sated in URM. For example, some endorsing journals had more lenient definitions of authorship, and also used different forms to declare COI. Interestingly, in our 2019 study, we also found that 1 journal did not have its ItA written by the journal editor(s) or publisher, but instead just pointed the authors to full ICMJE URM recommendations stating they apply in full. No other journal, that we are aware, does this.

DISCUSSION:

While the heterogeneity of the papers is well-noted, I think that there were more interesting themes that could have been extracted from the rich data. For example, why did 61% of studies not specify their analytic method?

Reply: We thank the reviewer for this comment, and as we said above, due to the plethora analysis we conducted, we tried to focus on the results of the meta-analyses. But we do mention our take on this particular question in the revised methods, in the section on study bias. Even though our study was not designed to answer why the authors did not report the methods or ItA years, we can assume this occurred for several reasons – first, we included studies from 1986, which is a time before reporting guidelines became common in biomedicine. Second, the “analytic method” in these papers largely consists of reading *Instruction to Authors*, and counting when and sometimes how a specific topic is covered. And it is very likely that when this was done by only a single person, authors felt there was no need to explain this. Finally, some researchers may consider it standard practice that data extraction and reading of documents are done by two independent individuals, and therefore consider there is no need to state it in the paper – however, we feel this to be less likely.

Minor errors throughout – I assume will be corrected in the editorial process:

First paragraph: lost) in line 4. “Conducting these studies” (no “of”). Increase in the number of studies” [not “of”]. Maybe add the word “gradual” to the claim about 2002? When were ItAs first online? Well before 2002, I would think, so the “switch to online publishing” as a cause might be more fully explored.

Second paragraph: “Although” not “Though”. Drop “the” between “that better reporting of”. I think that you should add a sentence recognizing the role of peer reviewers and editors in helping to ensure authors’ compliance with ItAs as well.

Paragraph 4: Confusing. “Probably corresponds to the improvement” “increased attention to research integrity.”

Reply: We thank the reviewer for catching these errors and we have corrected all of them in the revised manuscript. The answer to the growth of articles we gave above. Additionally, we expanded the discussion to recognize the role of the reviewers and editors in ensuring compliance with ItAs.

Why did no Health Sciences papers discuss URM? One concern is the vast representation of Health Sciences journals in the final set of studies analysed. Were there more comparisons that could be made within this subset of studies? Perhaps this is a separate paper that the team intends to publish; this would be of great interest.

Reply: We are not sure what the reviewer means with “no Health sciences discuss URM”. We stated in the results and in the appendix, that 45 studies looked if URM was endorsed (of which 41 looked at URM coverage in Health journals, and 4 in non-Health journals). Our appendix includes 5 pages (pp. 28 to 32) of the analysis of data just regarding URM. Perhaps the reviewer did not have access to our appendix. We feel we have conducted all comparisons for this subset of studies, and as we stated above, no other analyses are planned (nor we feel possible) with this dataset for the six research integrity topics.

Paragraph 6 – the suggestion of a database to compare publications’ ItAs is a wonderful idea. I hope that the team can help implement such a project. “for timely posting of clinical trial results”.

Reply: We thank the reviewer for the comment. It is our personal wish to create such a database, but until that happens, we are happy to inform we are also working on monitoring the outputs of all studies that sought ethics approval at my university, with the wish to expand it to all studies (even those not requiring that approval).

Paragraph 9 – While briefly mentioned, the concept that some authors will always be more rigorous than required by ItAs should be strengthened. ItAs are minimum standards and the most rigorous papers will often exceed the requirements stated, incorporating other standards and emerging best practices. ItAs should never preclude authors for adhering to better reporting.

Reply: We fully agree with the author, and have expanded this point in the revised manuscript.

METHODS

Information Sources and Search

The authors state that the full search strategies for all three databases are available on their project repository. I have scoured that site and downloaded the full data files and cannot find the search strategies. While I am not inclined to doubt the validity of the searches – librarian Ana Utrobicic is acknowledged – I find it troubling that the search strategies for any systematic review are so difficult to find and verify since the search strategy is the foundation of any systematic review. The text refers to searches done on 1 May 2017, which is over three year ago. While analyzing the volume of data for this project undoubtedly took a significant amount of time, I am concerned that the search was not updated before the manuscript was submitted. Considering that there were only 812 studies to screen originally, an update would not have taken long. Even a brief literature search shows that additional papers on this topic have been published, although I don’t know if they would have been included in the analysis because the authors don’t clearly state their inclusion or exclusion criteria in either the text or PRISMA flow

chart. Also, they do not appear to have subjected the included studies to any appraisal of quality. Overall, the systematic review aspects of the study could have been more successfully reported.

Reply: I sincerely apologize that the search strategies were not on the website. I should have double checked before submitting the manuscript was submitted. They can be found now on the website, and have been included at the end of this rebuttal letter. As the reviewer correctly guessed, the analysis of all this data, took us more than 18 months, and we are unable to update it at this time, as MM (the first author) has left the institution where this project started and the study funding has ended. Furthermore, even if only a few studies came out in the last year, they would require we update every single meta-analysis and sub-analysis we did, which would even further increase the 41 pages of our appendix, and require updating every table and figure we created. But even more importantly, is highly unlikely that adding these studies would change any of the main messages of the paper. Therefore, it is just not feasible for us at this point to do an update, and we hope the editor and reviewers will understand that. The revised version includes a clear statement about the timeframe and an adapted title, as suggested by the reviewer. Additionally, we would like to mention that the first manuscript (systematic review part) was submitted to the journal on February 2019, and on journal's recommendation we merged it with the meta-analysis and submitted in May 2020. Finally, on a personal note, as this study presents the 30 years of research studies of this topic (1987 to 2017) it is our wish to update it when 40 years have passed, and then, if we are lucky maybe even every 10 years after. Ideally if we are able to create a database of all ItA recommendations for all journals – a continuous tracking of these topics will also at that point be available for all researchers and publishers. Finally, the inclusion criteria and the reason for lack of appraisal of studies are now more clearly stated in the revised methods, where all the PRISMA items are now listed.

Conclusion:

Is there a role for a “lessons learned” section or advice for others who might want to replicate this study?

Reply: Per Nature Communications formatting requirements, no lessons learned section can be a part of the main manuscript, but we did include those lessons in our notes about data extraction and data preparation on our project's website. Each update of our database (5 so far) have all been followed with notes on how data was improved or cleaned. Additional analyses notes are also available in our appendix.

With their commitment to standards and reporting, I would hope that this team would also conform to the emerging standards for reporting searches for systematic reviews – PRISMA-S (<https://osf.io/ygn9w/>). A baseline study such as this one will be difficult to reproduce if the searches are not documented and available. Including the search strategy of at least one of the searches in the appendix published with the article would be preferable to using an ephemeral data repository.

Reply: Full search for all 3 databases is now included, per PRISMA-S recommendations. As stated above, I again apologize for the mistake of not checking that it was uploaded during the initial submission. Additionally, per PRISMA-S recommendations: “*authors should upload complete documentation to a data repository, an institutional repository, or other secure and permanent online archive instead of relying on journal publication*” and so the search is included on our project's website (Mendeley Data repository) rather than in appendix.

I am not trained to evaluate the statistical analysis of this paper.

I withhold judgement on the overall paper until I can review the search strategies. But I am likely to recommend the publication of this manuscript with minor revisions.
Holly Grossetta Nardini

Reply: We thank the reviewer for the comments, and hope that we have addressed all aspects that would allow for a full judgment on the paper.

Final note: We would like to thank the editor and the reviewers again for their comments, and we hope that you will find the revised manuscript suitable for publication in your journal.
In the name of the co-authors,
Mario Malički

Search strategies (performed on 1 May 2017)

Developed with the help of Ana Utrobičić (Central Medical Library, University of Split School of Medicine)

Databases:

Ovid MEDLINE(R) In-Process & Other Non-Indexed Citations and Ovid MEDLINE(R)
<1946 to Present>

Search Strategy:

- 1 (instruction* adj3 author*).tw. (2476)
- 2 (instruction* adj3 journal*).tw. (97)
- 3 (submi* adj3 guideline*).tw. (170)
- 4 1 or 2 or 3 (2679)
- 5 journals.tw. (28371)
- 6 4 and 5 (213)

WoS Strategy:

(TS=(instruction* NEAR/3 author*) OR TS=(instruction* NEAR/3 journal*) OR TS=(submi* NEAR/3 guideline*)) AND TS=(journals) NOT SO=(AESTHETIC PLASTIC SURGERY)

Results 317

Scopus:

(TITLE-ABS-KEY ((instruction* W/3 author) OR (instruction* W/3 journal*) OR (submi* W/3 guideline*))) AND (TITLE-ABS-KEY (journals)) AND NOT ABS ("See Instructions to Authors for a complete description") AND NOT ABS ("please refer to the Table of Contents or the online Instructions to Authors") AND NOT ABS ("Instructions for Authors for a complete description of levels of evidence") AND NOT ABS ("Guidelines for submission are available at") AND (EXCLUDE (EXACTSRCTITLE , "Aesthetic Plastic Surgery "))

Results 749

Reviewer #1 (Remarks to the Author):

I am happy with the revisions made in response to my comments, no other comments to add from my end.

Reviewer #2 (Remarks to the Author):

After reviewing the revised manuscript and response to referees letter I am satisfied that the authors have addressed my comments.

Reviewer #3 (Remarks to the Author):

Thank you for your detailed responses to my concerns about the original manuscript. You have answered my concerns fully, and I am especially interested in the additional table that you have prepared. I have reviewed the search strategies and am glad that they were fully reported.